# *VastTrack*: Vast Category Visual Object Tracking

Liang Peng[1*],  Junyuan Gao[1*],  Xinran Liu[1,3*],  Weihong Li[1,2,3*],  Shaohua Dong[4*],
Zhipeng Zhang[5],  Heng Fan[4†],  Libo Zhang[1,2,3†♯]

[1]Institute of Software Chinese Academy of Sciences  [2]Hangzhou Institute for Advanced Study
[3]University of Chinese Academy of Sciences  [4]University of North Texas  [5]KargoBot
[*]Equal contribution  [†]Equal advising and co-last authors  [♯]Corresponding author

## Abstract

In this paper, we propose a novel benchmark, named *VastTrack*, aiming to facilitate the development of general visual tracking via encompassing abundant classes and videos. VastTrack consists of a few attractive properties: **(1)** *Vast Object Category*. In particular, it covers targets from 2,115 categories, significantly surpassing object classes of existing popular benchmarks (*e.g.*, GOT-10k with 563 classes and LaSOT with 70 categories). Through providing such vast object classes, we expect to learn more general object tracking. **(2)** *Larger scale*. Compared with current benchmarks, VastTrack provides 50,610 videos with 4.2 million frames, which makes it to date the largest dataset in term of the number of videos, and hence could benefit training even more powerful visual trackers in the deep learning era. **(3)** *Rich Annotation*. Besides conventional bounding box annotations, VastTrack also provides linguistic descriptions with more than 50K sentences for the videos. Such rich annotations of VastTrack enable the development of both vision-only and vision-language tracking. In order to ensure precise annotation, each frame in the videos is manually labeled with multi-stage of careful inspections and refinements. To understand performance of existing trackers and to provide baselines for future comparison, we extensively evaluate 25 representative trackers. The results, not surprisingly, display significant drops compared to those on current datasets due to lack of abundant categories and videos from diverse scenarios for training, and more efforts are urgently required to improve general visual tracking. Our VastTrack, the toolkit, and evaluation results are publicly available at `https://github.com/HengLan/VastTrack`.

## 1  Introduction

Visual tracking is a fundamental computer vision problem with many applications such as surveillance and robotics. The ultimate goal for tacking is to localize the target of an *arbitrary* category in an *arbitrary* scenario from a sequence, given its initial position, which we term *universal visual object tracking*. For such goal, numerous trackers have been proposed in recent decades [58, 45, 34, 28, 39]. In particular, with the introduction of several large-scale tracking benchmarks (*e.g.*, [43, 16, 27]) in the deep learning era, considerable advancements (*e.g.*, [7, 9, 57, 5, 38, 6, 51, 37]) have been seen in the visual tracking community. Despite this, it remains challenging to achieve universal tracking.

One important reason is relatively *restricted* number of object categories in current tracking benchmarks. The objects in the real world are from *countless* categories. To achieve general visual tracking like humans, the tracker is expected to "SEE" various sequences from an extremely large set of object categories during training to acquire the generalization ability. Nevertheless, the categories in existing large-scale benchmarks are rather *limited*. For example, the popular TrackingNet [43] and LaSOT [16] comprise respectively 27 and 70 categories (see Fig. 1), which fall short for training universally generalizable trackers. Another popular dataset GOT-10k [27] aims to handle this by largely expanding the number of object categories to 563. Despite its success in advancing generic-purpose

tracking, the 563 object categories are still insufficient to represent massive diversity of categories present in the real world. Besides training, a real general tracking system requires evaluation on videos of vast object categories, which can help mitigate biases to certain classes for more faithful assessment in real applications. Nevertheless, the test sets of existing large-scale benchmarks (*e.g.*, [43, 16, 27]) all consist of *less than* 100 categories, which may not be enough for faithful assessment of general tracking.

Besides rich categories, abundant videos with high-quality annotations are crucial for learning robust visual trackers. Particularly, as a tracking model becomes larger and more complicated, *e.g.*, from CNNs [30, 26] to Transformer [47], more videos are demanded to unleash the power of deep network for achieving robustness and generality. While there have been extensive efforts to develop tracking datasets, they are comparatively small in scale or limited in annotation quality. For example, currently the largest (in term of video number) benchmark [27] with *precise* annotations has only 10K videos, which may still be inadequate for training generalized trackers, as evidenced by enhanced performance [9] on it when using extra training videos. Although another benchmark [43] offers more videos, its annotations are not precise, which may degrade performance.

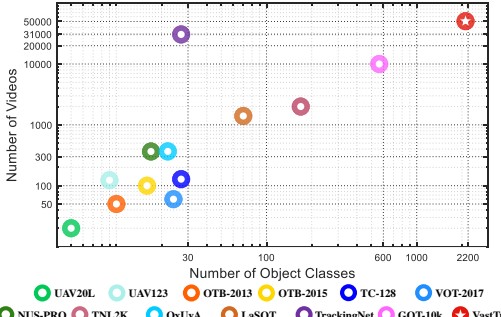

Figure 1: Summary of representative benchmarks, comprising OTB-2013/2015 [53, 54], TC-128 [36], UAV123 [42], NUS-PRO [32], UAV20L [42], VOT-2017 [29], OxUvA [46], GOT-10k [27], TrackingNet [43], and VastTrack. We can clearly see that VastTrack is *larger* than all other datasets by containing 2,115 object categories and 50,610 videos. *Best viewed in color for all figures.*

More recently, language has demonstrated great potential to enhance robustness of general tracking, and the resulted paradigm, the so-called *vision-language tracking* (*e.g.*, [35, 24, 19, 61]), has attracted increasing attention. For learning a robust and general vision-language tracker, it is crucial to provide ample videos with visual and linguistic annotations. Although there are several datasets (*e.g.*, [16, 50]) created for this goal, the number of linguistic sentences are limited in scale (*e.g.*, 1.4K in [16] and 2K in [50]), which may impede the exploration of more general vision-language tracking.

In order to alleviate the aforementioned limitations in existing datasets for developing more general visual tracking, we propose *VastTrack*, an innovative large-scale benchmark for **Vast**-category short-term object **Track**ing via comprising abundant classes and video from diverse scenarios. In particular, VastTrack makes the following efforts for facilitating the development of general object tracking:

**(1)** *Vast Object Category*: To enrich the diversity in object categories for general tracking, VastTrack consists of videos from 2,115 classes, which largely surpasses category number in popular benchmarks such as GOT-10k [27] with 563 classes and LaSOT [16] with 70 classes, as displayed in Fig. 1. To our best of our knowledge, VastTrack is the richest tracking dataset with the largest number of categories. With such vast object classes, we expect to accelerate the exploration towards more general tracking.

**(2)** *Larger Scale*: For learning robust universal tracking, our VastTrack offers 50,610 video sequences with 4.2 million frames, which makes it so far the largest and the most diverse tracking dataset in terms of the numbers of videos and targets compared to existing datasets (*e.g.* [16, 27, 43]), as shown in Fig. 1. Such a larger scale and diversity of VastTrack in videos and targets can potentially benefit training more powerful trackers, particularly Transformer-based models, in the deep learning era.

**(3)** *Rich and Precise Annotations*: Considering the benefits of language for enhancing general object tracking, VastTrack offers both standard bounding box annotations and rich linguistic specifications for the sequences, and thus enables exploration of both the vision-only and vision-language universal tracking. Compared with current benchmarks (*e.g.*, [16] with 1.4K and [50] with 2K sentences) for vision-language tracking, the proposed VastTrack provides over 50K descriptions, a magnitude order larger than [16, 50], of more and diverse targets for better vision-language tracking. In addition, to ensure precise annotations, each video in VastTrack is manually labeled with multi-round refinements.

In order to understand the performance of existing trackers on VastTrack and to provide baseline for future comparison, we extensively evaluate 25 recent representative algorithms in a hybrid protocol in which the test videos have partial overlap with the training sequences (as described later) and conduct

Table 1: Comparison of VastTrack with other datasets. "ST" and "LT" indicate short- and long-term tracking. "Tra." and "Eav." mean training and evaluation. ♭: TrackingNet [43] is semi-automatically labeled using a tracker. ¶: The number of object classes in TNL2K [50] is obtained by our statistic.

| Benchmark | Year | Classes | Videos | Mean Frames | Total Frames | Total Duration | Absent label | Num. of Att. | Lang. Anno. | Frame Rate | Dataset Focus | Dataset Goal |
|---|---|---|---|---|---|---|---|---|---|---|---|---|
| OTB-2013 [53] | 2013 | 10 | 50 | 578 | 29K | 16.4 min | ✗ | 11 | ✗ | 30 fps | ST | Eva. |
| OTB-2015 [54] | 2015 | 16 | 100 | 590 | 59K | 32.8 min | ✗ | 11 | ✗ | 30 fps | ST | Eva. |
| TC-128 [36] | 2015 | 27 | 128 | 429 | 55K | 30.7 min | ✗ | 11 | ✗ | 30 fps | ST | Eva. |
| NUS-PRO [32] | 2016 | 17 | 365 | 371 | 135K | 75.2 min | ✗ | 12 | ✗ | 30 fps | ST | Eva. |
| UAV123 [42] | 2016 | 9 | 123 | 915 | 113K | 62.5 min | ✗ | 12 | ✗ | 30 fps | ST | Eva. |
| UAV20L [42] | 2016 | 5 | 20 | 2,934 | 59K | 32.6 min | ✗ | 12 | ✗ | 30 fps | LT | Eva. |
| NfS [21] | 2017 | 17 | 100 | 3,830 | 383K | 26.6 min | ✗ | 9 | ✗ | 240 fps | ST | Eva. |
| VOT-2017 [29] | 2017 | 24 | 60 | 356 | 21K | 11.9 min | ✗ | 24 | ✗ | 30 fps | ST | Eva. |
| OxUvA [46] | 2018 | 22 | 366 | 4,235 | 1.55M | 14.4 hours | ✗ | 6 | ✗ | 30 fps | LT | Eva. |
| TrackingNet [43] | 2018 | 27 | 30,643 | 471 | 14.43M♭ | 140.0 hours | ✗ | 15 | ✗ | 30 fps | ST | Tra./Eva. |
| LaSOT [16] | 2019 | 70 | 1,400 | 2,053 | 3.52M | 32.5 hours | ✓ | 14 | ✓ | 30 fps | LT | Tra./Eva. |
| TNL2K [50] | 2021 | 169¶ | 2,000 | 622 | 1.24M | 11.5 hours | ✓ | 17 | ✓ | 30 fps | ST | Tra./Eva. |
| GOT-10k [27] | 2021 | 563 | 9,935 | 149 | 1.45M | 40.0 hours | ✓ | 6 | ✗ | 10 fps | ST | Tra./Eva. |
| **VastTrack** | 2024 | 2,115 | 50,610 | 83 | 4.20M | 194.4 hours | ✓ | 10 | ✓ | 6 fps | ST | Tra./Eva. |

in-depth analysis. The evaluation reveals that, not surprisingly, current top-performing object trackers significantly degrade on the more challenging VastTrack. For example, the success scores of existing state-of-the-art trackers, *e.g.*, SeqTrack [6], MixFormer [9], and OSTrack [57], degrade from 0.725, 0.724, and 0.711 on LaSOT [16] to 0.396 (with a drop of 0.329), 0.395 (with a drop of 0.329), and 0.336 (with a drop of 0.375) on VastTrack. This demonstrate the challenge in achieving universal tracking for current trackers, and more efforts are desired to improve general-purpose object tracking.

By releasing VastTrack, we expect to offer a new large-scale platform with abundant videos from vast categories for facilitating the development of more general and universal tracking and its applications. In summary, our main ***contributions*** are as follows: ♠ We introduce a new benchmark VastTrack that covers 2,115 object categories to facilitate more general tracking; ♥ VastTrack provides a large scale of 50,610 videos which could benefit developing more powerful deep trackers; ♣ Rich annotations in VastTrack enable exploration of both vision-only and vision-language tracking; ♦ Evaluation of 25 trackers is conducted to understand VastTrack and provides baselines for future comparison.

## 2 Related Work

**Visual Tracking Benchmarks.** Benchmarks have been crucial for development of tracking. Early tracking benchmarks are usually in small scale and mainly aim at the evaluation purpose for fairly comparing different algorithms. OTB-2013 [53] is the first tracking benchmark by introducing 51 videos and later extended in [54] by adding new sequences. VOT [29] presents a series of challenges to compare trackers in different aspects. TC-128 [36] contains 128 colorful videos to study the impact of color information on tracking models. NfS [21] assesses tracking performance by providing 100 sequences with high frame rate. UAV123 and UAV20L [42] respectively consist of 123 and 20 videos captured by unmanned aerial vehicle for tracking performance evaluation. NUS-PRO [32] offers 365 videos to assess trackers on rigid target objects. OxUvA [46] contains 366 sequences for evaluating long-term tracking performance of different algorithms. From a different perspective than opaque tracking benchmarks, TOTB [17] collects 225 videos for investigating transparent object tracking.

Despite facilitating tracking, early datasets are limited in scale and cannot provide videos for training deep tracking. To alleviate this, several large-scale benchmarks have been introduced in recent years. TrackingNet [43] presents a large-scale dataset with around 30K videos for training deep tracking. However, its annotations are generated using a tracker, which may be inaccurate and thus degrade the training of deep tracking. LaSOT [16] comprises 1,400 long-term videos with precise dense annotations, and is later extended in [15] by adding more videos. Notably, it provides both bounding box and language annotation to enable both vision-only and vision-language tracking. GOT-10 [27] contributes a large benchmark with around 10K sequences from 563 classes. Despite advancing deep tracking, 563 object classes may still be insufficient to represent massive categories in the real world.

VastTrack is related to the aforementioned large-scale datasets but provides a more diverse and larger platform with more than 50K videos from 2,115 categories, which aims to accelerate the exploration towards universal and general tracking. Tab. 1 shows the comparison of VastTrack with other datasets.

**Vision Benchmarks with Vast Categories.** Benchmarks with vast object categories are desired for learning general vision systems. Numerous such benchmarks have been introduced for various vision tasks. For example, the well-known ImageNet [14] consists of 1,000 classes for image recognition. Open Image [31] covers 600 categories for object detection. LVIS [25] comprises 1,203 classes for the tasks of object detection and instance segmentation. TAO [13] contains 833 categories for general multi-object tracking. The recently proposed V3Det [48] contributes a new dataset with 13,204 object classes with the goal of facilitating the general detection system development.

In the similar spirit with the above vast category benchmarks, we introduce VastTrack that comprises 2,115 object classes and more than 50K sequences for visual tracking. To the best of our knowledge, VastTrack is so far the largest tracking benchmark regarding the categories and videos and we hope it can serve as a cornerstone dataset for developing more general object tracking systems.

## 3 The Proposed VastTrack

### 3.1 Construction Principle

The goal of VastTrack is to develop a unique large-scale platform with abundant object categories and video sequences with rich as well as precision annotations for facilitating the development of more general tracking. For this purpose, we follow principles below in constructing our VastTrack:

- *Vast Object Category.* One key motivation of VastTrack is to facilitate more universal object tracking with a rich class diversity. To this end, we hope that the new benchmark covers at least 2,000 object classes, containing common target objects suitable for visual tracking in our life.
- *Larger Scale.* Abundant sequences are crucial for training deep trackers. We expect VastTrack to include at least 50K videos with an average video length of at least 80 frames. Such a scale, greatly larger than current datasets, can potentially benefit training more powerful deep trackers.
- *Rich Annotation.* One of important goals of VastTrack is to develop a comprehensive platform that supports both vision-only and vision-language tracking. Considering this, both bounding boxes and language specifications will be provided to boost tracking in different directions.
- *High Quality.* The quality of annotation is crucial for both training and evaluation. To ensure high quality of VastTrack, we manually label each video with multi-round inspections and refinements.

### 3.2 Data Acquisition

VastTrack aims to cover abundant categories for tracking. To this end, 2,115 categories are selected for building VastTrack. These object categories are chosen from different sources, including classes in ImageNet [14] and V3Det [48], WordNet [41], and Wikipedia, and organized in a hierarchical tree structure. Note that, each selected category is verified by an expert (*e.g.*, a PhD or MS student working on the related topic) to ensure that it is suitable for the tracking task. Compared with existing datasets, the object classes of VastTrack are more diverse and more desired for universal tracking as discussed before. Please refer to **supplementary material** for details of object classes in VastTrack.

After determining all object categories of VastTrack, we then search for the sequences of each class from YouTube under Creative Commons licence. The reason to use YouTube for sourcing videos is because it is currently the largest the video platform and many videos come from the real world. Initially, we gather more than 66K sequences. Then, we carefully inspect each video for the availability for visual tracking task, and finally pick out 50,610 sequences. For each qualified video, we remove the irrelevant content from it, and only retain an usable clip for tracking. Note that, unlike LaSOT [16] in

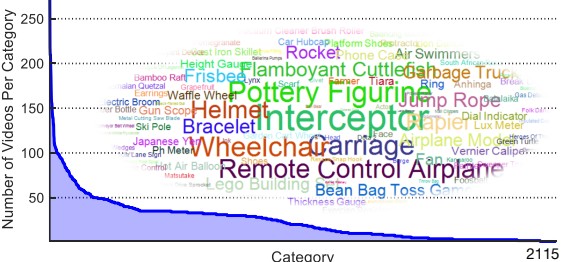

Figure 2: The number of videos in each object class forms a long-tail distribution, which is common and universal in our real world.

which each category has the same number of videos, the sequence number of each class is not equal, forming a long-tail distribution (see Fig. 2) that is more universal in real world and could encourage learning more practical and general visual trackers [27].

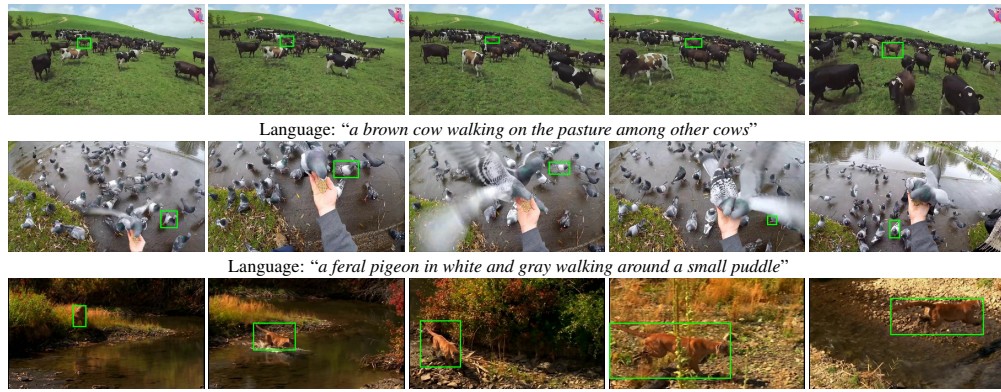

Language: "*a brown cow walking on the pasture among other cows*"

Language: "*a feral pigeon in white and gray walking around a small puddle*"

Language: "*a brown cougar crossing the river and running on the ground*"

Figure 3: Visualization of several annotation examples in the proposed VastTrack.

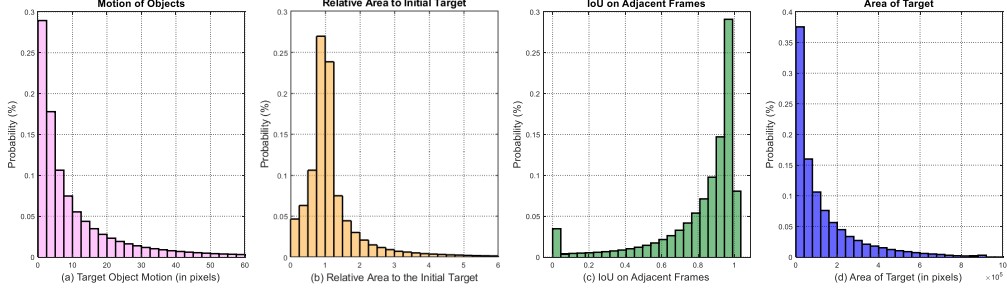

Figure 4: Statistics of annotations on object motion (image (a)), relative area compared to the initial object (image (b)), IoU of targets in adjacent frames (image (c)), and size of targets (image (d)).

Eventually, we develop a new benchmark, VastTrack, by covering 2,115 categories. It contains 50,610 videos with 4.2 million frames with an average sequence length of 83 frames. Because of limited space, we display detailed distribution of video length of VastTrack in the **supplementary material**. Please **notice** that, VastTrack is focused on short-term tracking by offering abundant object classes and sequences. Despite this, it can still be used for training long-term temporal trackers, as evidenced by the effectiveness of short-term videos in [27, 43] for learning robust trackers on both long-/short scenarios. In order words, diversity and quantity of objects and videos may be more crucial for deep tracking. It is worth **noting** that, although the frame rate (*i.e.*, 6 *fps* as in Tab. 1) of VastTrack is less than that of traditional benchmarks, this may not impact the training and evaluation of tracking much. Specifically, on the training side, since most of current tracking frameworks adopt the training mechanism of frame sampling with an interval (*e.g.*, 100 frames), our VastTrack can be used for training by choosing a suitable interval (probably less than the interval used for traditional datasets). On the evaluation side, according to the analysis in [46], labeling at different frame rate, even at 1 *fps*, does not adversely affect the robustness of tracking evaluation.

### 3.3 Annotation

We follow the similar principle as in [16, 15] for the bounding box annotation of a sequence: given the initial target object, in each frame, if the object shows up in the view, a labeler manually draws its (axis-aligned) bounding box as the tightest one to fit any visible part of the target; otherwise an absence label, either *out-of-view* or *full occlusion*, is given to the frame. Notice that, for some categories such as "*Kite*" and "*Yo-Yo*", the string does not belong to the target object to track, and thus will not be included in the annotated bounding box.

Guided by the above principle, we compile an annotation team with a few experts and a qualified labeling group, and adopt a multi-step mechanism, including manual labeling, visual inspection and refinement. In the first step, after experts label the initial target in the first frame, the annotation group starts to label the target in all other frames in the video. Notice that, to ensure consistency, each video

is labeled (and refined if necessary) by the same annotator. After this, in the second step, the experts verify the completed annotations from the first step. If the annotation is not unanimously agreed by a validation team (formed by two or three experts), it will be returned back to the original labeler for refinement in the third step. Throughout the annotation process, the second and third steps are repeated for multiple rounds, which ensures high-quality annotations of VastTrack. Fig. 3 displays several annotation examples. In Fig. 4, we show the distributions of target motion, relative area to the initial object, Intersection over Union (IoU) between targets in adjacent frames, and the size of object. From these statistics, we can see that objects moves fast and varies rapidly in the videos of VastTrack.

Considering the benefits of language in improving tracking (*e.g.* [35, 24, 19, 61]), we offer language specifications, besides box annotations, for videos in VastTrack, aiming to facilitate the development of vision-language tracking. In specific, a sentence of natural language that describes color information, behavior, and surroundings of the object as well as optionally its interaction with other objects is given as the linguistic annotation for the video (see Fig. 3 for examples). Although there have been datasets for similar goal (*e.g.* [16, 50] as in Tab. 1), the scale is limited by containing 1.4K [16] and 2K sentences [50]. Differently, VastTrack offers over 50K videos with richer linguistic specifications for different objects, and thus may benefit learning more powerful vision-language trackers.

### 3.4 Attributes

To enable further in-depth analysis, we offer ten attributes for *test* videos in VastTrack, including (1) invisibility (INV), assigned when object is partially or fully invisible due to occlusion or out of view, (2) deformation (DEF), assigned when target is deformable, (3) rotation (ROT), assigned when object rotates, (4) aspect ratio change (ARC), assigned when ratio of bounding box aspect ratio is outside [0.5, 2], (5) illumination variation (IV), assigned when illumination in object region heavily varies, (6) scale variation (SV), assigned when ratio of bounding box

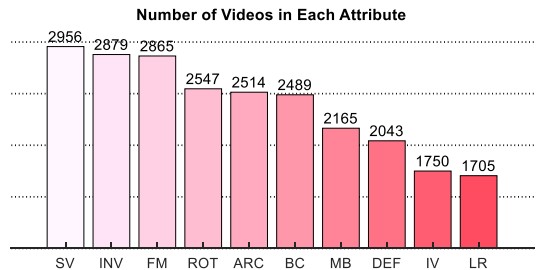

Figure 5: Distribution of videos per attribute.

is outside [0.5, 2], (7) fast motion (FM), assigned when target center moves larger than its size in last frame, (8) motion blur (MB), assigned when blur in object regions occurs (9) background clutter (BC), assigned when the similar appearance (not necessarily the same class of target) as target appears, and (10) low resolution (LR), assigned when target region is less than 1,000 pixels. For each video, a 10D binary vector is adopted to indicate the presence of an attribute, *i.e.*, "1" for presence , "0" otherwise.

The distribution of attributes for the test videos of VastTrack is shown in Fig. 5. We can see that the most common challenge is scale variation, involved with 2,956 videos. In addition, invisibility due to partial or full occlusion or out-of-view and fast motion frequently occur with 2,879 and 2,865 videos.

### 3.5 Dataset Split and Evaluation Protocol

**Dataset Split.** VastTrack has 50,610 videos, with 47,110 videos for training in VastTrack$_{Tra}$ and the rest 3,500 for testing in VastTrack$_{Tst}$. Tab. 2 displays comparison of training and testing sets. In dataset split, we try to keep distributions of training and testing sets similar. Please note, the reason to use 3,500 videos (∼7% of

Table 2: Comparison of *training* and *testing* sets.

|  | Classes | Videos | Mean frames | Total frames |
|---|---|---|---|---|
| VastTrack$_{Tst}$ | 702 | 3,500 | 106.3 | 372**K** |
| VastTrack$_{Tra}$ | 1,974 | 47,110 | 81.2 | 3.82**M** |

total) in VastTrack$_{Tst}$ is to keep it relatively compact so that evaluation of trackers can be fast, similar to the popular GOT-10k [27] in which 420 videos out of around 10K are for testing (∼4.2% of the total). Although VastTrack$_{Tst}$ has only 3,500 videos, it is representative by including rich categories and various scenarios for evaluation, and much larger and more diverse compared to other testing sets in video number and classes, making evaluation more reliable.

**Evaluation Protocol.** Unlike the *full overlap* [16, 43] or *one-shot* [27], we utilize a *hybrid* protocol wherein part of object classes (not videos) in test set have overlap with training set, while the rest classes remains unseen. The reason is that, in real world, humans often track objects from both

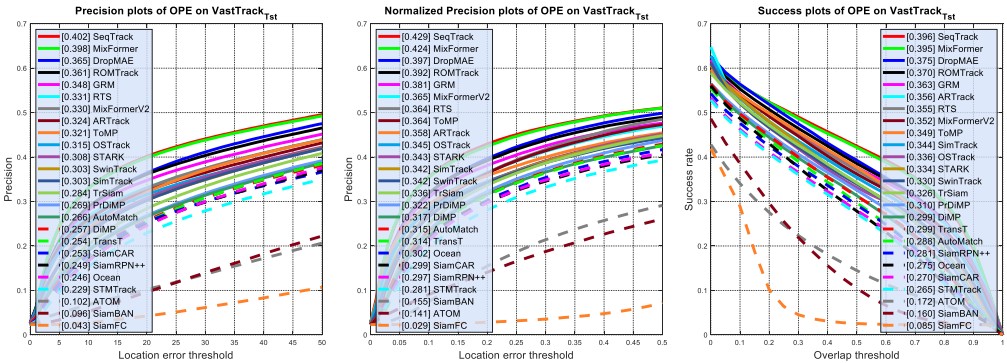

Figure 6: Evaluation results of 25 trackers on VastTrack_Tst using PRE, NPRE, and SUC.

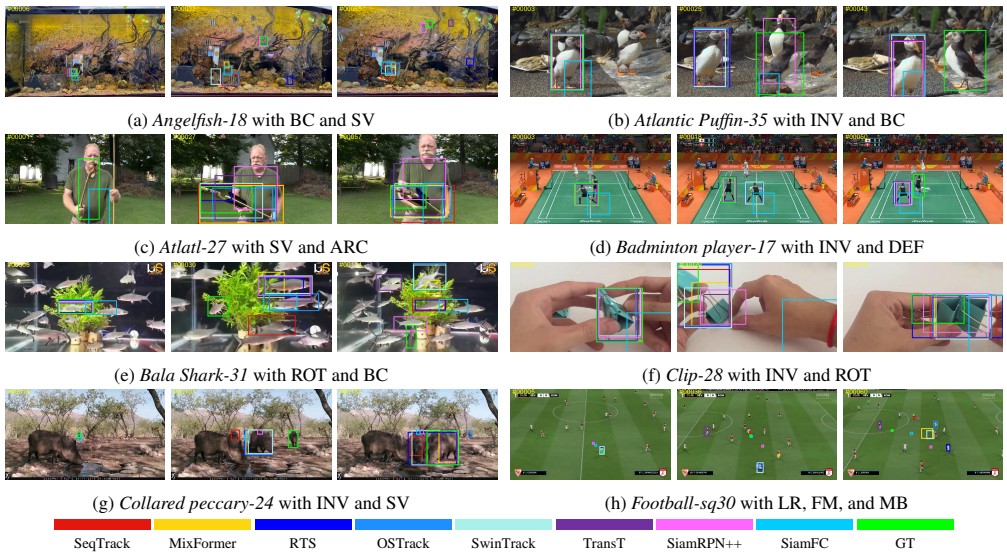

(a) *Angelfish-18* with BC and SV

(b) *Atlantic Puffin-35* with INV and BC

(c) *Atlatl-27* with SV and ARC

(d) *Badminton player-17* with INV and DEF

(e) *Bala Shark-31* with ROT and BC

(f) *Clip-28* with INV and ROT

(g) *Collared peccary-24* with INV and SV

(h) *Football-sq30* with LR, FM, and MB

Figure 7: Qualitative results of eight representative trackers on different sequences. We observe that these trackers drift to the background region or even lose the target due to different challenges in videos such as BC, SV, DEF, INV, MB, ROT, and LR. More efforts are desired to improve tracking.

frequently seen and unseen categories. To develop human-like trackers, we adopt such a hybrid protocol for VastTrack with 561 overlap classes and 141 completely unseen classes in test set.

## 4 Experiments

**Evaluation Metric.** Following [53, 16, 43], we use *one-pass evaluation* (OPE) and compare different trackers using three metrics, including *precision* (PRE), normalized precision (NPRE), and success (SUC). In specific, PRE measures center position distance between tracking results and groundtruth in pixels, and trackers are ranked by PRE on a preset threshold, *e.g.*, 20 pixels. To mitigate influence of video resolutions, NPRE is calculated by normalizing PRE using target region. Different from PRE and NPRE, SUC measures Intersection over Union (IoU) between tracking results and groundtruth, and is computed by the percentage of frames in which the IoU is larger than a threshold, *e.g.*, 0.5.

### 4.1 Evaluated Trackers

To understand existing approaches on VastTrack and also to offer baselines for comparison, we evaluate 25 representative trackers, which are classified into three types: **(i) CNN-based** that achieves object tracking using only CNN architecture, consisting of SiamFC [1], ATOM [11], SiamRPN++ [33], SiamBAN [8], DiMP [2], SiamCAR [23], PrDiMP [12], STMTrack [20], Ocean [60], RTS [44], and

AutoMatch [59]; **(ii) CNN-Transformer-based** that implements visual tracking via hybrid CNN and Transformer architectures, including STARK [56], TrSiam [49], TransT [7], and ToMP [40]; **(iii) Transformer-based** that tracks the target through leveraging a pure Transformer architecture. The tracking approaches in this category consist of OSTrack [57], SwinTrack [38], MixFormer [9] and MixFormerV2 [10], SimTrack [5], SeqTrack [6], ARTrack [51], DropMAE [52], and ROMTrack [4] as well as GRM [22]. We conduct the evaluations on a workstation with an Intel Xeon w9 CPU and 4 Nvidia A6000 GPUs. Please note, all trackers are assessed as they are, without modifications. A detailed summary of these trackers are shown in the **supplementary material** due to limited space.

We understand that tracking is a rapidly evolving field with numerous trackers proposed every year. To keep the evaluations on VastTrack as up-to-date as possible, we try our best to maintain lead board on our project page by continuously including newly published trackers (*e.g.*, [55, 3, 37]) from major avenues. Please refer to our project webpage for more details.

### 4.2   Evaluation Results

**Overall Performance.** We evaluate 25 trackers on VastTrack, including many recent Transformer-based methods. Note that, for evaluation, each tracker is evaluated as it is, without any modification. The evaluation results are reported in Fig. 6. We can see that, SeqTrack achieves the best performance on all three metrics with 0.402 PRE, 0.429 NPRE, and 0.396 SUC scores, MixFormer displays the second best results with 0.398 PRE, 0.424, and 0.395 SUC scores, and DropMAE obtains the third best results with 0.365 PRE, 0.397 NPRE, and 0.375 SUC scores. All these three trackers are developed based on vision Transformer architecture, showing its power in feature learning for tracking. Notably, although RTS does not employ Transformer architecture for tracking, it still achieves promising results with 0.331 PRE, 0.364 NPRE, and 0.355 SUC scores, even better than a few Transformer trackers like OSTrack with 0.315 PRE, 0.345 NPRE, and 0.336 SUC scores and SwinTrack with 0.303 PRE, 0.342 NPRE, and 0.330 SUC scores. We argue this is because RTS adopts tracking-by-segmentation which is beneficial for tracking object with extreme aspect ratio. Note that, the recent MixFormerV2 with 0.330 PRE, 0.365 NPRE, and 0.352 SUC scores performs worse than its previous version MixFormer, because it leverages much lighter network for efficiency. An interesting observation is that, SiamRPM++, a seminal Siamese tracker, surprisingly outperforms many its extensions such as SiamCAR, Ocean, and SiamBAN, showing its generality to some extent.

**Qualitative Evaluation.** In addition to the quantitative evaluation in the main text and this appendix, we further show qualitative results on VastTrack. Specifically, we demonstrate visualizations of eight representative trackers, including SeqTrack, MixFormer, RTS, OSTrack, SwinTrack, TransT, SimaRPN++, and SiamFC in different attributes such as *scale variation*, *deformation*, *rotation*, *aspect ratio change*, *background clutter*, *invisibility*, *blur*, *fast motion*, and *low resolution* in Fig. 7. As displayed Fig. 7, we can observe that, although the trackers can deal with some challenges in the video sequences, they may still fail in more complicated scenarios where multiple challenges occur simultaneously, which indicates that more efforts are desired to improve existing approaches towards universal visual tracking.

**Discussion.** The evaluation shows some useful observations: (1) *Feature network*. As shown in Fig. 6, we observe that, the top five trackers are based on Vision Transformer architecture, which reveals that the exploration of more powerful feature network is still an important direction for improving tracking. This is consistent with findings in other benchmarks. Despite adopting powerful feature network, the performance is still far from satisfaction, compared to that on other benchmarks (as shown later). We argue this is caused by the lack of universal large-scale training of more general object categories for tracking. (2) *Temporal information*. Videos contain abundant temporal information which is important for tracking. However, this is largely ignored to some extend owing to the great success of Siamese tracking in recent years. Especially, even without using temporal information, many trackers still achieve state-of-the-art performance. However, from Fig. 6, we see that, the top three trackers all leverage temporal information for tracking, which indicates the crucial role of temporal cue for tracking. We hope, through evaluation results on VastTrack, researchers can pay more attention in developing robust tracking by incorporating temporal cues.

**Attribute-based Performance.** To better analyze different trackers, we further perform attribute-based evaluation on ten challenges. Fig. 8 shows the attribute-based results for the two most common attributes of SV and INV (based on number of videos in each attribute) and two difficult attributes of LR and IV using SUC. As in Fig. 8 (a), SeqTrack, MixFormer, and DropMAE achieves the best three

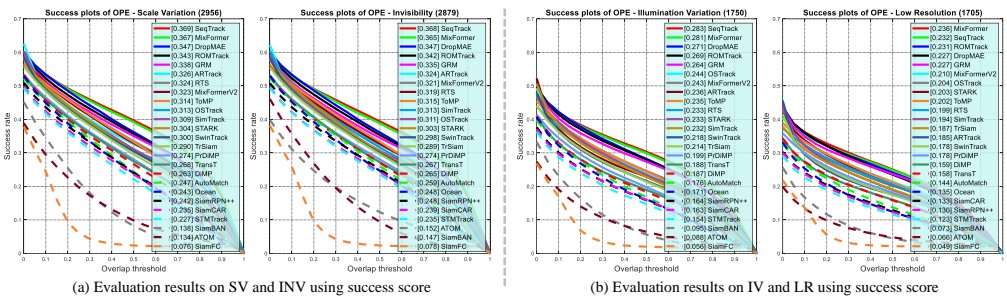

(a) Evaluation results on SV and INV using success score   (b) Evaluation results on IV and LR using success score

Figure 8: Evaluation on the two most common attributes (a) and difficult attributes (b) using SUC.

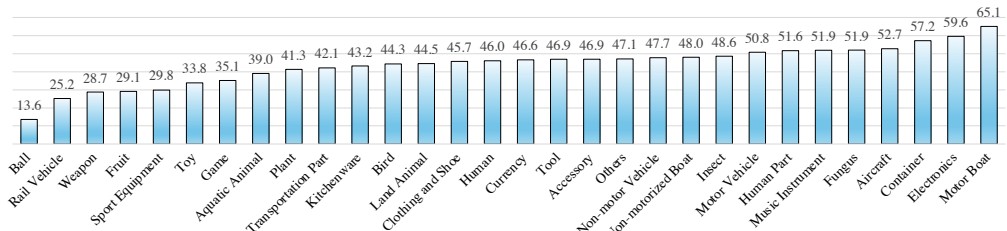

Figure 9: Comparison on different meta categories using SUC score.

results on SV/INV with 0.369/0.368, 0.367/0.365, and 0.347/0.347 scores in SUC, which is consistent with their performance in overall evaluation. As in Fig. 8 (b), SeqTrack, MixFormer, and DropMAT are the best three trackers on IV. An interesting finding is, IV is considered to be easy for tracking [18]. Nevertheless, our result shows difference. We argue, this is because IV in VastTrack usually occurs in low-light condition with complicated background, which degrades tracking performance. This also shows, extreme illumination change still need to be carefully dealt with. LR is the most difficult challenge in VastTrack, because it may result in low-quality feature extraction. On LR, MixFormer, SeqTrack, and ROMTrack achieve the best three results with 0.236, 0.232, and 0.231 SUC scores.

**Comparison on Meta Category.** Fig. 9 compares different meta categories (or coarse classes in the **supplementary material**) using $SUC_{meta}$, calculated using SUC scores of all trackers on a certain meta class. From Fig. 9, we can see that, some common categories such as boat, aircraft, vehicle and human parts are relatively easy for tracking, while rare classes like weapon, fruit, and various balls are hard to locate probably due to the lack of enough training videos from these categories, which indicates the need of more object categories for learning general tracking systems.

Due to space limitation, more evaluation and analysis can be seen in the **supplementary material**.

### 4.3 Comparison to Other Benchmarks

Compared to existing benchmarks, the proposed VastTrack is more challenging due to the requirement of tracking object from more classes (in test). We present a comparison of VastTrack and other popular large-scale tracking benchmarks including TrackingNet [43], LaSOT [16], and TNL2K [50]. Please note that, GOT-10k [27] here is not compared because it adopts different metrics for evaluation. Tab. 3 reports the results of the top 15 trackers on our VastTrack and their results on TrackingNet, LaSOT, and TNL2K using SUC. From Tab. 3, we observe that, all the compared trackers have a heavy performance drop on VastTrack. For

Table 3: Comparison to other datasets.

| | Success Score | | | |
|---|---|---|---|---|
| | TrackingNet [43] | LaSOT [16] | TNL2K [50] | VastTrack (Ours) |
| SeqTrack [6] | 0.855 | 0.725 | 0.578 | 0.396 |
| MixFormer [9] | 0.854 | 0.724 | 0.533 | 0.395 |
| DropMAE [52] | 0.841 | 0.718 | 0.569 | 0.375 |
| ROMTrack [4] | 0.841 | 0.714 | 0.604 | 0.370 |
| GRM [22] | 0.840 | 0.699 | 0.611 | 0.363 |
| ARTrack [51] | 0.843 | 0.708 | 0.575 | 0.356 |
| RTS [44] | 0.816 | 0.697 | 0.599 | 0.355 |
| MixFormerV2 [10] | 0.834 | 0.706 | 0.506 | 0.352 |
| ToMP [40] | 0.815 | 0.685 | 0.584 | 0.349 |
| SimTrack [5] | 0.834 | 0.705 | 0.556 | 0.344 |
| OSTrack [57] | 0.839 | 0.711 | 0.559 | 0.336 |
| STARK [56] | 0.820 | 0.671 | 0.525 | 0.334 |
| SwinTrack [38] | 0.811 | 0.672 | 0.559 | 0.330 |
| TrSiam [49] | 0.781 | 0.624 | 0.523 | 0.326 |
| PrDiMP [12] | 0.758 | 0.598 | 0.470 | 0.310 |

instance, SeqTrack, the best tracker on VastTrack, achieves high SUC scores of 0.855, 0.725, and 0.578 on TrackingNet, LaSOT, and TNL2K, while degrades to 0.396 on VastTrack with 0.459, 0.329, and 0.182 drops. OSTrack drops from 0.839, 0.711, and 0.559 SUC scores on TrackingNet, LaSOT, and TNL2K to 0.336 on VastTrack. SwinTrack degrades from 0.811, 0.672, and 0.559 on the existing benchmarks to 0.330 on VastTrack. Likewise, other trackers suffer similar drops, which shows the challenge for current trackers and there is still a long way for improving tracking.

In addition, we see an interesting observation about the relative performance of trackers from Tab. 3. Specifically, we see that a few trackers such as OSTrack and SimTrack performing better on LaSOT may perform relatively worse than others such as GRM, RTS, and ToMP on VastTrack. We argue that the possible reason is the abilities of different trackers in dealing with overfitting for object categories, which shows the need of more diverse videos with different classes in learning more general tracking.

## 4.4 Retraining Experiments with VastTrack

In order to demonstrate the effectiveness of VastTrack in improving existing methods, we retrain two trackers, consisting of SiamRPN++ [33] and OSTrack [57], using the training set of VastTrack through fine-tuning. Tab. 4 displays the results. As in Tab. 4, we clearly see that, after further training, the SUC score of SiamRPN++ is clearly improved from 0.281 to 0.298 with performance gains of 1.7% on VastTrack,

Table 4: Further training with VastTrack.

|  | SiamRPN++ [33] | | OSTrack [57] | |
|---|---|---|---|---|
|  | SUC w/o retraining | SUC w/ retraining | SUC w/o retraining | SUC w/ retraining |
| VastTrack | 0.281 | 0.298 (↑1.7%) | 0.336 | 0.362 (↑2.6%) |
| LaSOT | 0.496 | 0.528 (↑3.2%) | 0.711 | 0.722 (↑1.1%) |
| TrackingNet | 0.733 | 0.762 (↑2.9%) | 0.839 | 0.852 (↑1.3%) |
| TNL2K | 0.413 | 0.446 (↑3.3%) | 0.559 | 0.579 (↑2.0%) |

from 0.496 to 0.528 with 3.2% gains on LaSOT, from 0.733 to 0.762 with 2.9% gains on TrackingNet, and from 0.413 to 0.446 with 3.3% gains on TNL2K. Besides, for OSTrack, the SUC score is boosted from 0.336 to 0.362 with 2.6% improvements on VastTrack, from 0.711 to 0.722 with 1.1% gains on LaSOT, from 0.839 to 0.852 with 1.3% gains on TrackingNet, and form 0.559 to 0.579 with 2.0% gains on TNL2K. Note that, OSTrack is already strong on LaSOT but still enhanced using VastTrack. All these experiments validate the effectiveness of VastTrack in improving tracking. In the future, we will further explore the potential of VastTrack for advancing tracking performance.

## 5 Conclusion

In this paper, we propose a novel large-scale benchmark, dubbed VastTrack, to facilitate the development of more general object tracking. To this goal, VastTrack contains abundant object categories and video sequences. Specifically, it covers 2,115 classes and 50,610 videos with 4.2 million frames. To the best of our knowledge, VastTrack is to date the largest benchmark regarding class diversity and video number. Besides, VastTrack offers both bounding box annotations and language descriptions, which enables exploring both vision-only and vision-language tracking. In order to ensure the high quality, VastTrack is manually annotated with multi-round of careful inspection and refinement. We evaluate 25 trackers to analyze existing methods on VastTrack and to offer baselines for comparison. The evaluation results reveal that more efforts are needed for general tracking. By releasing VastTrack, we expect to provide a cornerstone dataset for developing more general object tracking.

**Limitation.** Despite the vast object categories and larger scale of our VastTrack, there are limitations. First, given the proposed large-scale VastTrack with vast object classes, a baseline that outperforms other trackers is not provided. Second, since most video sequences in VastTrack are relative short, it may not be suitable for long-term tracking performance evaluation, though it can be used for training long-term temporal trackers. Considering that our primary goal in this work is to offer a new benchmark with vast object categories, we leave these questions to further work by designing new trackers and by developing a new subset for long-term tracking evaluation.

**Social Impact.** By releasing VastTrack, researchers from the tracking community are able to leverage it as a platform for training more general tracking systems as well as for assessing and comparing different models, which could facilitate the deployment of tracking in more real-world applications.

**Acknowledgment.** Libo Zhang was supported by National Natural Science Foundation of China (No. 62476266). Heng Fan was not supported by any fund for this work.

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

# A  Appendix

In this appendix, we provide more details, analysis, results, and discussion of VastTrack (project with data, code, and results is at `https://github.com/HengLan/VastTrack`), including

- **A1 Details of Object Classes**
  We show the detailed object classes in VastTrack and number of sequences in each category.
- **A2 Statistic of the Video Length**
  We present the statistic of video length on VastTrack.
- **A3 Summary of Evaluated Trackers**
  We show the detailed summary of 25 evaluated trackers.
- **A4 Full Results of Attribute-based Evaluation**
  We show the full attribute-based evaluation results for all evaluated trackers in precision (PRE), normalized precision (NPRE), and success (SUC).
- **A5 Maintenance, Ethical Issue, and Responsible Usage of VastTrack**
  We discuss the maintenance, ethical issue, and responsible usage of our VastTrack.

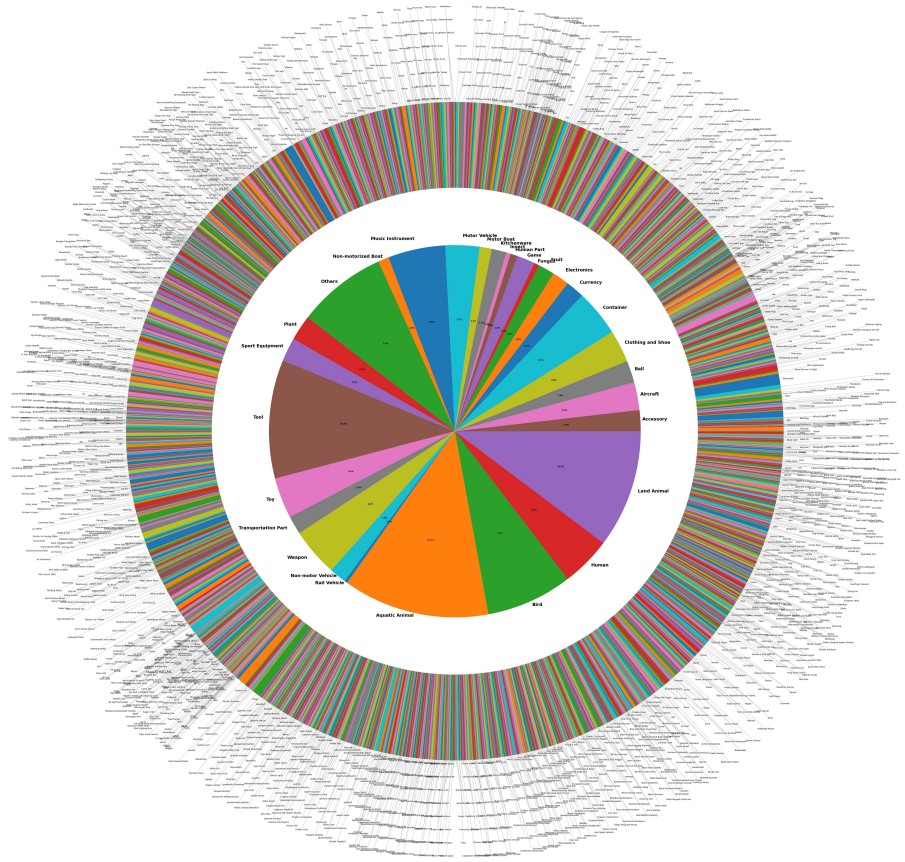

Figure 10: Category organization of VastTrack. Please zoom in.

## A.1  Details of Object Classes

VastTrack covers 2,115 classes, aiming to facilitate the development of universal and general object tracking. We collect these categories in a hierarchical way. In specific, we first collect 30 coarse classes, including "*Human*", "*Human Part*", "*Land Animal*", "*Aquatic Animal*", "*Bird*", "*Motor Vehicle*", "*Non-motor Vehicle*", "*Motor Boat*", "*Non-motorized Boat*", "*Accessory*", "*Aircraft*", "*Ball*", "*Clothing and Shoe*", "*Container*", "*Currency*", "*Electronics*", "*Fruit*", "*Kitchenware*", "*Game*", "*Insect*", "*Fungus*", "*Music Instrument*", "*Plant*", "*Sport Equipment*", "*Tool*", "*Toy*", "*Weapon*", "*Transportation Part*", "*Rail Vehicle*", and "*Others*". Then, we further gather 2,115 fine object categories in each coarse classes. All these fine categories are verified by the expert to ensure that

the sequences in this class are suitable for tracking. Fig. 10 displays the category organization in VastTrack. In order to enable better understanding of VastTrack, below we present each object category and the number of sequences in the format of "*Class* (# videos)", *e.g.*, "*Cow* (35)", that means object category of "Cow" with 35 videos, in the alphabetical order as follows:

## A (107 classes)

*Ardwolf* (18), *Accessory Box* (21), *Accordion* (33), *Acipenser Sinensis* (13), *Acrobat* (35), *Actinomycetes* (4), *Activity Wall* (3), *Actor* (82), *Adhesive Tape* (58), *Adjustable Ladder* (3), *Adjustable Wrench* (10), *Aerial Drones In Flight* (50), *Aerial Silk Performers* (48), *Aerial Work Platform* (34), *Afghan Hound* (20), *Afm* (20), *African Skimmer* (4), *Aim* (3), *Aims* (3), *Aiptasia Eating Filefish* (35), *Air Ambulance* (31), *Air Hockey* (30), *Air Swimmers* (119), *Aircraft Carrier* (33), *Airedale Terrier* (3), *Airline Stewardess* (15), *Airplane Model* (141), *Airport Vehicle* (25), *Airportshuttle* (27), *Airship* (35), *Ais74* (3), *Ak100* (3), *Ak102* (3), *Ak103* (3), *Ak104* (3), *Ak105* (3), *Ak47* (3), *Ak74M* (3), *Akm* (3), *Akm63* (3), *Akms* (3), *Aks74* (3), *Aks74U* (3), *Alaskan Malamute* (3), *Alligator* (31), *Almond* (5), *Aloe Vera* (2), *Aloe Vera Cactus* (7), *Alpaca* (20), *Ambulance* (35), *Amd65* (3), *American Bulldog* (20), *American Cocker Spaniel* (3), *American Eskimo Dog* (3), *Amphibious Vehicle* (21), *Amulet* (26), *An94* (3), *Anchor Winch Wheel* (33), *Andean Cock-Of-The-Rock* (18), *Andinoacara Latifrons* (36), *Angelfish* (35), *Angle Gauge* (33), *Angle Grinder* (2), *Anhinga* (100), *Ankle Socks* (3), *Ankle Boots* (15), *Ankylosaurus* (9), *Annona* (6), *Antarctic Shag* (5), *Antarctic Tern* (10), *Antelope* (44), *Anti-Radiation Clothing* (3), *Apistogramma Eremnopyge* (47), *Apollo Shark* (8), *Apothecary Box* (20), *Ar15* (3), *Arctic Hare* (24), *Arctic Warbler* (31), *Arctic Wolf* (21), *Arena Of Valor* (32), *Argentine Peso* (15), *Arm Warmers* (3), *Armadillo* (33), *Armored Searobin* (7), *Armoured Personnel Carrier* (11), *Art Supply Box* (20), *Arterial Line Kit* (1), *As50* (1), *Asian Arowana* (28), *Asian Lady Beetle* (30), *Atlantic Puffin* (44), *Atlatl* (54), *Audio* (26), *Audio Box* (7), *Audio Equipment Caster* (10), *Auger* (3), *Auricularia Auricula* (35), *Australian Cattle Dog* (2), *Australian Dollar* (15), *Australian Pratincole* (10), *Autoinjector* (3), *Aviator Glasses* (3), *Avocado* (14), *Awaous Flavus* (14), *Azaras Night Monkey* (3), *Azarass Capuchin* (2), *Azure-Winged Magpie* (31).

## B (226 classes)

*Baboon* (57), *Baby Bottle* (26), *Backhoe Loader* (53), *Backpack* (55), *Badis Ruber* (8), *Badminton* (30), *Badminton Player* (69), *Badminton Racket* (30), *Bagh Nakh* (2), *Baglama* (32), *Bala Shark* (8), *Balalaika* (93), *Balance Pods* (10), *Balance Wheel* (34), *Balancing Machine* (88), *Bald Uakari* (4), *Ballerina Pumps* (74), *Ballet Dancer* (39), *Ballpoint Pen* (47), *Ballpoint Pen Fish* (32), *Bamboo Raft* (94), *Bamboo Rat* (32), *Banana* (29), *Band Saw* (20), *Bandage* (28), *Banggai Cardinalfish* (35), *Banjo* (29), *Bank Vole* (35), *Barbadian Dollar* (58), *Barbat* (33), *Bare-Eared Squirrel Monkey* (5), *Barge* (77), *Barley* (5), *Barn Swallow* (30), *Barred Cuckoo-Dove* (35), *Barrel Cactus* (10), *Barrel Jellyfish* (20), *Barrel Wheel* (68), *Baseball* (30), *Baseball Bat Swinging* (14), *Baseball Cap* (35), *Basketball* (30), *Basketball Player* (29), *Basketball Shoes* (55), *Basking Shark* (11), *Bass Clarinet* (11), *Bass Drum* (35), *Bass Saxophone* (63), *Basset Hound* (33), *Bat* (58), *Baton* (33), *Baton Twirling* (18), *Batteries Box* (53), *Battleaxe* (6), *Bayonet* (33), *Bcm4 Recce-14 Mcmr* (3), *Beach Pants* (4), *Beach Volleyball* (33), *Bean Bag Toss Game* (131), *Beanie Hat* (30), *Bearded Axe* (32), *Bearded Collie* (3), *Beaver* (40), *Bedlington Terrier* (3), *Bee* (29), *Beer* (29), *Beer Bottle* (93), *Belgian Malinois* (3), *Belizean Dollar* (11), *Belly Dancer* (28), *Belt* (4), *Belt Bag* (61), *Beluga Whale* (32), *Bench Grinder* (3), *Bench Swing* (3), *Bench Vise* (20), *Benelli Lupo* (3), *Beretta Holding M9A3* (11), *Beretta Holding 90Two* (4), *Beretta Holding 92F* (7), *Beretta Holding Apx* (6), *Bergara Bmr* (3), *Berthier Mle 1890* (3), *Betta Fish* (35), *Bhutan Takin* (28), *Bible* (63), *Bichon Frise* (19), *Bicycle Lane Sign* (83), *Bicycle Race Bike* (1), *Bifocal Glasses* (2), *Bigfin Reef Squid* (35), *Bike* (64), *Billiards* (30), *Binturong* (21), *Bird Wrasse* (35), *Birgus Latro* (11), *Black Ant* (13), *Black Bean* (8), *Black Bearded Saki* (4), *Black Caiman* (13), *Black Guillemot* (5), *Black Molly* (11), *Black Muntjac* (33), *Black Musk Deer* (4), *Black Panther* (25), *Black Skimmer* (5), *Black Swan* (34), *Black-Bellied Tern* (10), *Black-Fronted Titi Monkey* (1), *Black-Headed Gull* (60), *Black-Legged Kittiwake* (4), *Black-Necked Swan* (33), *Black-Tailed Gull* (71), *Black-Throated Bushtit* (33), *Black-Winged Pratincole* (9), *Blanket Octopus* (25), *Bleach Bottle* (32), *Blond Capuchin* (3), *Blood Administration Set* (3), *Blood Collection Needle* (3), *Blood Collection Set* (10), *Blood Collection Tube* (10), *Blood Draw Syringe* (5), *Blood Transfer Device* (7), *Blood Tube Holder* (10), *Blue Blubber Jellyfish* (34), *Blue Crab* (20), *Blue Picardy Spaniel* (3), *Blue Rock Thrush* (34), *Blue Sea Dragon* (16), *Blue Sheep* (34), *Blue Star Leopard Wrasse* (35), *Blue Whale* (8), *Blue-And-White Flycatcher* (71), *Blueberry* (19), *Blue-Crowned Hanging-Parrot* (30), *Blue-Footed Booby* (10), *Blue-Ringed Octopus* (30), *Bluethroat* (28), *Bluetooth Speaker* (74), *Blunt Cannula* (10), *Blunt Fill Needle* (3), *Bo Staff* (30), *Bo Staff Techniques* (44), *Boat Shoes* (15), *Bobcat* (35), *Bobsledder* (18), *Bobtail Squid* (18), *Bocce Ball* (29), *Bocce Ball Set* (35), *Bodhran* (35), *Bodianus Sepiacaudus* (3), *Body Wash Bottle* (34), *Bolas* (10), *Bolero Jacket* (3), *Bolt Cutter* (9), *Bonapartes Gull* (6), *Bongos* (31), *Bonobo* (31), *Book Bag* (24), *Boomerang* (30), *Boots* (4), *Border Collie* (2), *Boston Terrier* (3), *Bottle Cap* (38), *Bottle Opener* (29), *Bounce And Spin* (4), *Bouncing Bumpers* (10), *Bouncing Platforms* (1), *Bow And Arrow* (6), *Bowed Psaltery* (70), *Bowl* (35), *Boxer* (50), *Boxer Dog* (3), *Boxfish* (35), *Bra* (8), *Bracelet* (155), *Brad Nailer* (20), *Brahminy Blind Snake* (3), *Brazil Nut* (11), *Bread Machine* (29), *Break Dancer* (95), *Brick Tongs* (20), *Bridge Ices Before Road Sign* (6), *Briefcase* (16), *British Shorthair Cat* (5), *Brl* (28), *Bronze-Winged Courser* (10), *Broom* (33), *Brothers In Arms* (30), *Brown Bear* (32), *Brown Booby* (10), *Brown Howler* (5), *Brown Noddy* (10), *Brown Spider Monkey* (5), *Brown-Flanked Bush Warbler* (19), *Browning X-Bolt* (3), *Brown-Mantled Tamarin* (3), *Brush* (26), *Brussels Griffon* (2), *Bubble Machine* (22), *Bubble Play Area* (1), *Bubble Snail* (18), *Bucket* (34), *Bucket Backpack* (13), *Buckwheat* (8), *Buffy Saki* (3), *Buffy-Headed Marmoset* (17), *Bugle* (34), *Bull Terrier* (3), *Bulldozer* (34), *Bullock Cart* (35), *Bumper Boats* (10), *Bumper Car* (6), *Bungee Trampoline* (10), *Bunker Tanker* (53), *Bunny Ears Cactus* (6), *Burmese Ferret-Badger* (20), *Butis Butis Fish* (9), *Butterfly* (29), *Butterfly Needle* (3), *Butterfly Sword* (16), *Butterflyfish* (28), *Buzuq* (29).

## C (208 classes)

*Cable Cutter* (3), *Cajon* (35), *Cake Pan* (11), *Calendar* (4), *California Spangled Cat* (4), *Call Of Duty* (10), *Camel* (31), *Camera Backpack* (21), *Camera Wheel* (16), *Camisole* (4), *Camogie* (32), *Can Opener* (20), *Candy Box* (36), *Candy Crab* (13), *Canik Tp9* (9), *Cannonball Jellyfish* (39), *Cannula* (3), *Cape* (3), *Car Hubcap* (99), *Carcano M1891* (2), *Cardigan* (4), *Cargo Pants* (2), *Cargo Plane* (4), *Cargo Ship* (52), *Cargo Truck* (34), *Carnation* (26), *Carp* (32), *Carpenters Flasher Wrasse* (11), *Carpet Cleaner* (29), *Carpet Shark* (17), *Carriage* (209), *Carrot* (3), *Cart Caster Wheel* (42), *Cashew* (14), *Caspian Tern* (10), *Cassins Auklet* (2), *Cast Iron Skillet* (100), *Castor Bean* (7), *Cat* (45), *Catamaran* (45), *Catheter Insertion Tray* (1), *Catheter Introducer* (2), *Catheter Securement Device* (3), *Catheter Tip Syringe* (3), *Cattle Egret* (26), *Caulking Gun* (10), *Cavaquinho* (28), *C-Clamp* (22), *Celesta* (27), *Cell* (6), *Central American Spider Monkey* (4), *Central Venous Catheter* (3), *Cereal Box* (32), *Chain Drive Sprocket* (79), *Chain Sickle* (3), *Chakram* (1), *Chalk* (11), *Chalk Line* (10), *Chalkboard Eraser* (12), *Chambered Nautilus* (24), *Chandelier* (29), *Channa Argus* (31), *Channa Striata* (49), *Charger Loading Lee-Metford* (3), *Checkers* (30), *Cheese Box* (15), *Cheetah* (32), *Chef Bag* (11), *Cherry Shrimp* (35), *Chestnut* (8), *Chestnut-Bellied Rock Thrush* (21), *Cheytacm200* (3), *Chicken* (30), *Chickpea* (7), *Chiffon Skirt* (3), *Chigiriki* (4), *Chilean Skua* (5), *Chimaera* (30), *Chimpanzee* (35), *Chinese Algae Eater* (35), *Chinese Army Chess* (1), *Chinese Bulbul* (65), *Chinese Chess* (15), *Chinese Chongqing Dog* (20), *Chinese Cobra* (3), *Chinese High-Fin Banded Shark* (34), *Chinese Li Hua Cat* (5), *Chinese White-Browed Bird Warbler* (21), *Chitarra Battente* (35), *Chive* (2), *Chocolate Box* (24), *Chop Saw* (1), *Chopping Board* (18), *Chow Chow* (16), *Christmas Cactus* (8), *Chrysanthemum* (22), *Cicada* (20), *Cicada Killer Wasp* (30), *Cigar Box* (32), *Cirrhilabrus Naokoae* (10), *Civet* (80), *Claret Cup Cactus* (1), *Clarinet* (40), *Clavichord* (1), *Cleaning Cloth* (34), *Cleaning Gloves* (45), *Cleaning Paste* (34), *Clear Tote* (40), *Cleistocactus* (29), *Clerk* (1), *Climbing Boulders* (4), *Climbing Dome* (3), *Climbing Rope* (16), *Clip* (6), *Cloth* (47), *Clothing Package Box* (31), *Clown Killifish* (35), *Clown Loach* (14), *Clownfish* (32), *Coast Guard Cutter* (21), *Coat* (3), *Cockatoo Squid* (2), *Cocktails* (30), *Cocoa Bean* (7), *Coconut* (3), *Coconut Octopus* (35), *Coffee Bean* (7), *Coffee Box* (12), *Coffee Press* (14), *Coffinfish* (18), *Colander* (18), *Cold Planer* (3), *Collared Crow* (70), *Collared Peccary* (49), *Colombian Red Howler* (4), *Colombian White-Faced Capuchin* (3), *Colored Socks* (3), *Colorimeter* (71), *Command Conquer* (31), *Common Goldeneye* (35), *Common Kingfisher* (32), *Common Moorhen* (44), *Common Murre* (3), *Common Redstart* (31), *Common Treeshrew* (33), *Common Woolly Monkey* (3), *Commuter Bag* (40), *Compass* (35), *Computer Chair*

*Caster Wheel* (29), *Concrete Mixer Truck* (40), *Concrete Pump Truck* (55), *Concrete Spreader* (3), *Conditioner Bottle* (5), *Congas* (31), *Congo Tetra* (35), *Connector Caps* (3), *Construction Ahead Sign* (16), *Construction Workers* (48), *Contrabassoon* (35), *Convertible* (29), *Conveyor Belt Wheel* (72), *Coordinate Measuring Machine* (9), *Coracle* (48), *Coral Catshark* (31), *Cordless Handheld Power Scrubber* (35), *Cordless Screwdriver* (20), *Cordless Skipping Rope Reviews* (4), *Corn* (1), *Cornet* (32), *Cornflower* (33), *Corsac Fox* (35), *Corydoras* (35), *Cosmetic Gift Box* (36), *Cosmetics Case* (29), *Cotton-Top Tamarin* (3), *Cougar* (32), *Courteney Stalking Rifle* (3), *Cow* (35), *Cow Shark* (5), *Cownose Ray* (10), *Crab-Plover* (10), *Crane* (32), *Crane Barge* (47), *Crane Wheel* (10), *Crayon* (27), *Crested Auklet* (14), *Crested Ibis* (5), *Cricket* (4), *Cricket Bat* (3), *Croquet Set* (35), *Cross Line Laser* (15), *Crossbody Backpack* (49), *Crossbow* (49), *Crossbow Pistol* (32), *Crowbar* (10), *Cruise Ship* (47), *Ctenopoma Acutirostre* (49), *Cucumber* (2), *Cufflinks* (18), *Cup* (32), *Cupping* (35), *Curler* (20), *Curling* (39), *Curling Iron* (22), *Cyclist* (50), *Cz P11C* (10), *Cz76* (10), *Cz-Usa Model 557 Eclipse* (3).

## D (74 classes)

*Dagger* (5), *Dahlia* (33), *Daisy* (31), *Dalmatian* (3), *Damselfly* (3), *Danbau* (32), *Dandelion Siphonophore* (7), *Daniel Defense Delta 5 Pro* (3), *Dartboard Set* (34), *Daurian Hedgehog* (34), *Deer Crossing Sign* (23), *Depth Gauge* (35), *Desk Lamp* (30), *Detour Sign* (27), *Dhole* (30), *Dial Indicator* (105), *Dialysis Needle* (1), *Diamond Drill* (3), *Diamond Tail Flasher Wrasse* (33), *Dice* (7), *Didgeridoo* (35), *Digging Toys* (3), *Digital Multi-Meter* (34), *Digital Piano* (27), *Dilruba* (35), *Dinnerware* (4), *Dione Ratsnake* (3), *Disc Swing* (10), *Discus Fish* (35), *Disinfecting Cotton Balls* (31), *Diver* (33), *Diving Support Vessel* (36), *Djembe* (30), *Doberman Pinscher* (3), *Doctor* (1), *Document Box* (8), *Dogfish Shark* (15), *Dolly Wheel* (58), *Dolphin Gull* (5), *Dominican Peso* (27), *Domra* (35), *Dota 2* (81), *Double Bass* (35), *Double Bridge Glasses* (10), *Doweling Jig* (19), *Down Jacket* (3), *Dragon Dance* (35), *Dragon Dance Staff* (30), *Dragon Wrasse* (64), *Drainage Truck* (35), *Dress* (4), *Drift Boat* (47), *Drillship* (46), *Drop Slide* (3), *Drum Set* (13), *Drum Spinner* (7), *Drum Stick* (30), *Drywall Stilts* (20), *Dsc* (11), *Duck* (49), *Duct Tape* (1), *Duduk* (35), *Duffle Backpack* (4), *Dulcimer* (5), *Dumbo Octopus* (35), *Dune* (10), *Durian* (47), *Dusky Shark* (10), *Dust Mitt* (51), *Dust Mop* (29), *Duster* (30), *Dustpan* (33), *Dwarf Chin Cactus* (4), *Dwarf Gourami* (44).

## E (65 classes)

*Ear* (40), *Earrings* (94), *Eastern Caribbean Dollar* (12), *Ecuadorian Sucre* (32), *Edamame* (10), *Edelweiss* (5), *Egg* (15), *Egg Slicer* (10), *Eggbeater* (9), *Eggplant* (3), *Elden Ring* (35), *Electric Air Blower* (48), *Electric Air Hockey Table* (8), *Electric Air Purifier* (30), *Electric Animal Rides* (10), *Electric Broom* (98), *Electric Car* (38), *Electric Drill Driver* (35), *Electric Hedge Trimmer* (68), *Electric Lawn Edger* (29), *Electric Mixer* (6), *Electric Piano* (34), *Electric Power Scrubber Brush* (30), *Electric Power Washer Foam Cannon* (32), *Electric Ray* (10), *Electric Scooters* (10), *Electric Scrubber Machine* (28), *Electric Squeegee* (41), *Electric Ultrasonic Toothbrush* (69), *Electronic Product Box* (61), *Electronic Scale* (30), *Elegant Tern* (10), *Elephant* (37), *Elevator Pulley* (7), *Elliptical Machine Flywheel* (37), *Emperor Tamarin* (2), *Endlers Livebearers* (8), *English Horn* (31), *Enterprise* (19), *Epidural Kit* (4), *Epinephelus Marginatus* (44), *Eraser* (21), *Escrima Sticks* (8), *Espadrilles* (34), *Euphonium* (20), *Eurasian Badger* (19), *Eurasian Curlew* (33), *Eurasian Eagle-Owl* (33), *Eurasian Hoopoe* (5), *Eurasian Sparrowhawk* (30), *Eurasian Tree Sparrow* (33), *Euro* (39), *European Bee-Eaters* (28), *European Otter* (44), *European Polecat* (18), *European Robin* (21), *Exercise Resistance Band Wheel* (93), *Exercise Wheel* (48), *Exit Sign* (30), *Exotic Shorthair Cat* (5), *Extension Set* (3), *Extension Tubing* (3), *Eye Mask* (3), *Eyebrow* (19), *Eyes* (45).

## F (97 classes)

*F90* (3), *Face* (94), *Fake Collar* (4), *Falcata* (16), *Falcated Duck* (31), *Falling Rocks Sign* (8), *Fallow Deer* (30), *Famas* (3), *Fan* (132), *Fan Wh* (7), *Farm Tractor Wheel* (50), *Farmer* (99), *Fashion Glasses* (2), *Feder* (30), *Fedora Hat* (46), *Fencer* (67), *Fennec Fox* (49), *Feral Pigeons* (24), *Ferris Wheel* (32), *Ferris Wheel Axle* (55), *Ferris Wheel Support Structure* (51), *Ferrule Fitting* (75), *Ferry* (11), *Fiddler Crab* (4), *Fiddler Ray* (60), *Fidget Spinner* (9), *Field Hockey* (29), *Fighter Jet* (56), *Figt* (6), *Filipino Martial Arts Stick And Knife Techniques* (30), *Finless Porpoise* (13), *Finnish Mark* (2), *Fire Station Sign* (9), *Fire Truck* (40), *Fireboat* (20), *Firefighters* (35), *Firefighting Aircraft* (34), *Firemouth Cichlid* (21), *Fish Tank* (20), *Fishermen* (33), *Fishing Reel Spool* (74), *Fishing Vessel* (46), *Fishnet Stockings* (3), *Fistball* (32), *Fistula Needle* (2), *Fitness Roller* (73), *Flamboyant Cuttlefish* (158), *Flame Jelly* (9), *Flammulina Velutipes* (45), *Flapjack Octopus* (17), *Flea* (4), *Floating Board* (35), *Floating Crane* (32), *Floating Lily Pads* (8), *Floating Logs* (5), *Floating Stepping Stones* (1), *Floatplane* (59), *Floor Lamp* (6), *Floor Squeegee* (5), *Floral Pants* (4), *Flounder* (32), *Fluid Warming Set* (9), *Flush Syringe* (1), *Flute* (35), *Flying Ball* (25), *Flying Car* (54), *Flying Fish* (8), *Flying Fox* (15), *Flying Gurnard* (35), *Flying Kites* (36), *Flying Lemurs* (14), *Fn M1935* (10), *Fn509* (10), *Fn57* (9), *Folk Dancer* (79), *Food Tongs* (29), *Food-Print Shirt* (3), *Foosball* (96), *Foot* (42), *Football* (15), *Football Boots* (48), *Force Gauge* (14), *Fork* (10), *Fork-Tailed Sunbird* (40), *Formula Two* (49), *Forsters Tern* (9), *Foxtail Millet* (3), *Franchi* (3), *Fried Egg Jellyfish* (35), *Frisbee* (148), *Frog* (35), *Fruit Bat* (35), *Fruit Box* (1), *Fruit Fly* (5), *Frying Pan* (10), *Fur Coat* (4), *Futsal* (32).

## G (108 classes)

*G3* (3), *G36* (2), *Gaff Rig* (35), *Game Box* (1), *Ganoderma* (35), *Gansu Zokor* (6), *Garbage Bag* (25), *Garbage Can* (33), *Garbage Truck* (140), *Garden Cart Wheel* (96), *Garden Spider* (5), *Garden Warbler* (31), *Garlic* (4), *Garlic Press* (10), *Garment Bag* (8), *Gas Detector* (83), *Gastraphetes* (9), *Gaudy Clown Crab* (2), *Gavialis Gangeticus* (10), *Gc* (18), *Gemore* (29), *Geoffroys Tamarin* (2), *German Blue Ram* (35), *Ghost Crab* (35), *Ghost Pipefish* (38), *Giant Anteater* (48), *Giant Armadillo* (45), *Giant Australian Cuttlefish* (55), *Giant Inflatable Slides* (4), *Giant Otter* (33), *Giant Panda* (34), *Giant Saguaro Cactus* (35), *Giant Twister Game* (2), *Gibbon* (33), *Gift Box* (40), *Giraffe* (29), *Giri-Giri* (40), *Glaive* (2), *Glass Catfish* (16), *Glasses* (4), *Glassware* (10), *Glider* (43), *Glock G17* (6), *Glock G19X* (58), *Glock G23* (50), *Glock G45* (50), *Glockenspiel* (15), *Gloomy Nudibranch* (34), *Gloomy Octopus* (38), *Glue* (27), *Go* (2), *Goatfish* (35), *Goblin Shark* (2), *Goeldis Marmoset* (3), *Go-Karts* (10), *Gold-Dotted Flatworm* (26), *Golden Barrel Cactus* (32), *Golden Bush-Robin* (14), *Golden Lion Tamarin* (3), *Golden Pheasant* (35), *Golden Eagle* (19), *Golden-Bellied Capuchin* (5), *Golden-Handed Tamarin* (2), *Golden-Mantled Tamarin* (3), *Golf* (30), *Golf Player* (1), *Goliath Tigerfish* (64), *Googly-Eyed Stubby Squid* (10), *Gorilla* (39), *Grainsorghum* (3), *Grapefruit* (38), *Grapsus Grapsus* (45), *Grater* (9), *Gray Fox* (34), *Gray Leaf Monkey* (7), *Gray Whale* (25), *Great Bustard* (28), *Great White Shark* (16), *Great Egret* (25), *Greater Crested Tern* (9), *Green Cochoa* (28), *Green Leafhopper* (5), *Green Turtle* (88), *Greenland Shark* (8), *Grevys Zebra* (47), *Grey Jay* (32), *Grey Seal* (18), *Grey Treepie* (34), *Grey-Capped Greenfinch* (35), *Grey-Crowned Warbler* (12), *Grill Pan* (23), *Grizzly Bear* (30), *Ground Trampoline* (2), *Grouper* (49), *Grubfish* (18), *Guatemalan Quetzal* (89), *Guianan Squirrel Monkey* (4), *Guinea Pig* (26), *Guisarme* (3), *Guitar* (41), *Gull-Billed Tern* (10), *Gulper Eel* (13), *Gun Scope* (110), *Gymnast* (68), *Gymnocalycium* (49), *Gypsy Moth* (2), *Gyro Spinner* (10), *Gyroscope* (79).

## H (101 classes)

*Hacksaw* (6), *Hair Accessory Box* (11), *Hair Dryer* (16), *Hair Gel Bottle* (27), *Hairpin* (20), *Haitian Gourde* (53), *Half Banded Flasher Wrasse* (5), *Half-Life* (5), *Half-Moon Spear* (2), *Halichoeres Leucoxanthus* (15), *Halloween Crab* (18), *Halo* (10), *Hamidashi* (7), *Hammer Drill* (2), *Hammock Swing* (10), *Hand* (41), *Hand Pedal Bike* (13), *Handball Player* (39), *Handheld Game Console* (50), *Handkerchief* (4), *Hand-Powered Ferris Wheel* (5), *Hardness Tester* (26), *Hardware Box* (1), *Hare* (35), *Harlequin Rasboras* (11), *Harlequin Shark* (5), *Harlequin Shrimp* (31), *Harlequin Sweetlips* (35), *Harlequin Tuskfish* (35), *Harmonica* (14), *Harp* (17), *Harp Guitar* (34), *Harpoon* (25), *Harpsichord* (31), *Harvester* (31), *Hat* (33), *Hat Storage Box* (2), *Hawkthorn* (1), *Hazelnut* (9), *Head* (83), *Headband* (4), *Heavy-Lift Helicopter* (56), *Hedgehog* (32), *Hedgehog Cactus* (10), *Height Gauge* (108), *Helicopter* (34), *Helicopter Rotor* (32), *Helmet* (186), *Helmet Jellyfish* (18), *Helter Skelter Slide* (1), *Hemostasis Valve* (3), *Heparin Lock* (2), *Hericium Erinaceus* (33), *Heroes Of The Storm* (79), *Hibiscus* (31), *High Heels* (4), *High-Waisted Pants* (4), *Hiking Boots* (33), *Hill Prinia* (30), *Himalayan Monal* (32), *Hippopotamus* (35), *Historical Reenactments With Blades* (30), *Hk G28* (3), *Hk*

*Mk23 Mod 0* (4), *Hk Usp45* (5), *Hk Vp9* (4), *Hk416* (3), *Hockey Player* (24), *Hog Badger* (32), *Holacanthus Tricolor* (46), *Honey Bottle* (12), *Hooded Merganser* (44), *Hoodie* (4), *Hopper Balls* (10), *Horizontal Bar* (9), *Horned Puffin* (5), *Horse* (49), *Horse Mackerel* (8), *Horsebean* (8), *Hospital Sign* (20), *Hot Air Balloon* (103), *Hot Chocolate* (28), *Housefly* (20), *Hovercraft* (43), *Howa Stalker* (3), *Huber Needle* (2), *Hudson H9* (10), *Hula Hoop* (1), *Hula Skirt Siphonophore* (8), *Humpback Whale* (30), *Humphead Parrotfish* (35), *Hundred-Pace Viper* (4), *Hurdy-Gurdy* (34), *Hurling* (14), *Hwando* (4), *Hwandudaedo* (2), *Hydration Backpack* (28), *Hydraulic Breaker* (2), *Hydrometer* (6), *Hyena* (27), *Hypodermic Needle* (3).

## I (62 classes)

*Ia-2* (3), *Ibex* (39), *Ice Bag* (32), *Ice Cream Box* (5), *Ice Hockey Player* (3), *Ice Rescue Suit* (24), *Ice Speed Skating* (30), *Ice Skater* (35), *Icp* (11), *Imi Desert Eagle* (10), *Impact Driver* (19), *Impatiens* (29), *Imperial Shag* (23), *Inca Tern* (10), *Incense Burner* (27), *Inclinometer* (6), *Indian Ringneck Parrot* (31), *Indochinese Green Magpie* (35), *Indonesian Rupiah* (54), *Inflatable Bounce Houses* (7), *Inflatable Castle* (3), *Inflatable Obstacle Course* (5), *Inflatable Rescue Boat* (24), *Infrared Thermometer* (63), *Infusion Bags* (20), *Infusion Pump* (3), *Ink* (84), *Inline Hockey* (30), *Insect Repellent Bottle* (20), *Instrument Tray* (16), *Insulated Backpack* (29), *Insulated Cup* (76), *Insulin Pen* (3), *Insulin Pump* (3), *Interactive Fountains* (14), *Interactive Light Walls* (9), *Interactive Play Panel* (1), *Interactive Sound Play* (3), *Interceptor* (275), *Intermediate Egret* (42), *International Chess* (9), *Intradermal Needle* (3), *Intramuscular Needle* (3), *Intraosseous Needle* (3), *Iridescent Shark* (30), *Iron Gate Wheel* (13), *Iv Access Kit* (1), *Iv Armboard* (3), *Iv Ball* (3), *Iv Clamp* (5), *Iv Dressing* (3), *Iv Drip Chamber* (3), *Iv Flow Regulator* (9), *Iv Injector* (1), *Iv Pole* (2), *Iv Pressure Bag* (3), *Iv Push Syringe* (3), *Iv Site Protector* (3), *Iv Spike* (2), *Iv Tubing* (3), *Iv Warmer* (4), *Ivory Gull* (5).

## J (27 classes)

*Jack Dempsey Cichlid* (35), *Jackal* (27), *Jacket* (3), *Jackfruit* (50), *Jai Alai* (3), *Jam Jar* (28), *Jamaican Dollar* (35), *Japanese Bobtail Cat* (3), *Japanese Chin* (3), *Japanese Keelback* (3), *Japanese Macaque* (15), *Japanese Murrelet* (3), *Japanese Spitz* (3), *Japanese White Eye* (35), *Japanese Yen* (103), *Javelin* (25), *Jeans* (4), *Jerboa* (2), *Jerdon S Pitviper* (3), *Jewelry Box* (23), *Jigsaw Puzzle* (40), *John Dory* (35), *Judo Player* (35), *Juggling Clubs* (15), *Juice* (32), *Jump Rope* (156), *Jungle Gym* (3).

## K (30 classes)

*K2* (3), *Kangaroo* (80), *Karate Athlete* (27), *Kayak* (43), *Keep Off Median Sign* (2), *Kelp Gull* (10), *Keyboard* (24), *Kh2002* (1), *Kids Playing In Parks* (25), *Killer Whale* (35), *Kimber Hunter Pro Desolve Blak* (3), *Kin-Ball* (30), *Kinetic Ball* (1), *King Cobra* (2), *Kinkajou* (28), *Kissing Gourami* (33), *Kitchen Scissors* (10), *Kknd* (33), *Knee Pads* (4), *Knight* (25), *Knitwear* (4), *Knives* (10), *Knobkierrie* (6), *Koala* (29), *Korfball* (30), *Kpinga* (10), *Kuhli Loach* (32), *Kukri* (34), *Kung Fu Fan Dance* (15), *Kusarigama* (21).

## L (85 classes)

*L85A2* (2), *Labrador Retriever* (19), *Labrisomus Nuchipinnis* (3), *Labrisomus Philippii* (2), *Lacewing* (3), *Lacrosse* (10), *Ladyfinger Cactus* (4), *Laminate Trimmer* (1), *Lantern* (29), *Laptop* (50), *Large-Billed Crow* (34), *Large-Billed Tern* (10), *Large-Tailed Nightjar* (34), *Laser Distance Measurer* (57), *Laser Level* (48), *Latin Dancer* (70), *Laundry Bag* (37), *Lava Gull* (2), *Lavender* (2), *Lawn Bowling Set* (40), *Lawnmower Blade* (78), *Lawnmower Wheel* (32), *Lce Hockey* (23), *League Of Legends* (77), *Least Auklet* (5), *Lebel Mle 1886* (3), *Leggings* (4), *Lego Building Set* (138), *Legwarmers* (4), *Lemming* (10), *Lemon* (32), *Lemon Cactus* (4), *Lemon Shark* (29), *Lemur* (51), *Lentil* (1), *Leopard* (37), *Leopard Bush Fish* (41), *Lepomis Auratus* (31), *Lepomis Gulosus* (9), *Lepomis Macrochirus* (32), *Lepomis Marginatus* (17), *Lesser Frigatebird* (10), *Lesser Spotted Woodpecker* (45), *Lesser Panda* (33), *Lhasa Apso* (3), *Life Jacket* (46), *Lifebuoy* (16), *Lifting Hook Pulley* (20), *Light Rail Train* (34), *Lighter* (37), *Lilac* (32), *Linemans Pliers* (20), *Liobagrus Reinii* (1), *Lion Dance* (98), *Lionfish* (29), *Lipstick* (18), *Litchi* (43), *Little Auk* (2), *Little Egret* (52), *Little Forktail* (35), *Little Gull* (4), *Little Tern* (9), *Little Wart Snake* (15), *Lntravenous Catheter* (2), *Logistics Delivery Vehicle* (35), *Long Sword* (31), *Longan* (41), *Longear Sunfish* (28), *Longicorn Beetle* (5), *Longnose Hawkfish* (35), *Long-Tailed Goral* (2), *Long-Tailed Skua* (10), *Loofah* (4), *Loquat* (43), *Lotus Boat* (12), *Lucifer Titi Monkey* (5), *Luer Adapter* (2), *Luer Lock Cap* (7), *Luer Slip Syringe* (2), *Luger P08* (44), *Luggage Cart Wheel* (39), *Lunch Box* (24), *Lutjanus Decussatus* (2), *Lux Meter* (106), *Lynx* (87).

## M (133 classes)

*M1 Grand* (3), *M107* (3), *M110* (3), *M16* (3), *M1903* (3), *M1911A1* (3), *M1917* (2), *M2 Heavy Machine Gun* (3), *M25* (2), *M3* (2), *M40A1* (2), *M45A1* (10), *M4A1* (3), *M700* (3), *Macadamia Nut* (2), *Mace Spinning* (14), *Mackerel* (5), *Magazine Lee-Metford* (2), *Magic Mirror Maze* (2), *Magic Track* (60), *Magicians Performing Quick Hand Tricks* (3), *Maglev Train* (34), *Magnetic Sweeper* (3), *Magnificent Frigatebird* (9), *Magnifier* (1), *Mail Truck* (10), *Maine Coon Cat* (4), *Mainland Serow* (8), *Malayan Tapir* (49), *Malaysian Ringgit* (24), *Mallard* (21), *Maltese* (13), *Mammillaria* (35), *Manatee* (1), *Mandarin Duck* (35), *Mandarin Rat Snake* (3), *Mandarinfish* (35), *Mandocello* (35), *Mandolin* (3), *Mango* (37), *Mangosteen* (16), *Mangual* (12), *Mantis Shrimp* (8), *Mantled Howler* (2), *Many-Banded Krait* (3), *Marathon Runner* (51), *Marble Run Set* (28), *Marbled Cat* (3), *Marbled Hatchetfish* (35), *Marbled Murrelet* (12), *Marimba* (31), *Mascara Wand* (18), *Masked Booby* (10), *Mason S Square* (2), *Masonry Bit* (2), *Matsutake* (84), *Mattock* (33), *Mauve Stinger* (34), *Mbe* (11), *Mccoskers Flasher Wrasse* (11), *Measuring Cups And Spoons* (9), *Measuring Spoon* (9), *Meat Grinder Blade* (22), *Meat Slicer* (6), *Medical Cart Wheel* (57), *Medicine Bottle* (42), *Medieval Quarterstaff Combat* (27), *Melocactus* (34), *Merge Right Sign* (15), *Metal Cutting Saw Blade* (78), *Metal Lathe Chuck* (39), *Meteor Hammer* (25), *Microcontroller Tester* (19), *Micrometer* (24), *Micropterus Coosae* (22), *Micropterus Salmoides* (32), *Microscope* (34), *Midi Skirt* (3), *Military Transport Aircraft* (53), *Millers Saki* (4), *Mimic Octopus* (35), *Mini Bowling Alley* (8), *Mini Quadcopter* (32), *Mini Roller Coaster* (3), *Mini Soccer Field* (2), *Miniature Schnauzer* (19), *Miniature Village* (4), *Minimalist Wallet* (31), *Mink* (28), *Mirror* (33), *Mirrored Sunglasses* (5), *Miter Saw* (20), *Mixing Bowls* (7), *Mixing Syringe* (2), *Mk 14 Ebr* (3), *Mk12* (2), *Mk14* (3), *Mk20* (3), *Modern Martial Arts With Bo Staff* (20), *Modular Backpack* (19), *Moisture Meter* (29), *Mole Rat* (13), *Momentum Elite* (3), *Mongolian Jird* (16), *Monkey* (34), *Monkey Terrier* (3), *Monowheel* (31), *Moon Jellyfish* (35), *Moonlight Gourami* (30), *Moose* (33), *Morel* (34), *Morus Capensis* (3), *Mosasaurus* (38), *Mosin-Nagant M1891* (1), *Moth Bean* (5), *Motorcycle* (14), *Mountain Goat* (33), *Mountain Hare* (33), *Mp40* (3), *Mp5* (3), *Mp7* (3), *Mpi Km* (3), *Mpi Kms74* (3), *Mrad* (3), *Muffin Tin* (5), *Multirole Fighter* (29), *Mung Bean* (6), *Music Box* (30), *Music Express* (12), *Musk Deer* (10), *Musk Ox* (33), *Muskrat* (28), *Mustard Bottle* (50).

## N (32 classes)

*Nail Clippers* (73), *Nandao* (30), *Napo Saki* (3), *Narrow Bridge Sign* (4), *Nassau Grouper* (30), *Nature Observation Deck* (2), *Navanax* (4), *Neck Warmer* (3), *Needle Counter* (3), *Needle Disposal Container* (3), *Needle Holder* (3), *Needle Nose Pliers* (20), *Needleless Connector* (3), *Nepalese Rupee* (5), *New Zealand Dollar* (49), *Newt* (50), *Ney* (44), *Night Blooming Cereus* (32), *Niviventer Fulvescens* (3), *No Bicycles Sign* (1), *No Parking Fire Lane Sign* (25), *No Parking Sign* (24), *No Passing Zone Sign* (2), *No Turn On Red Sign* (32), *Northern Gannet* (9), *Northern Lapwing* (39), *Norwegian Krone* (25), *Notebook* (31), *Notebook Packing Box* (36), *Numbat* (42), *Nunchucks Play* (26), *Nyckelharpa* (33).

## O (29 classes)

*Oats* (2), *Oboe* (42), *Observation Aircraft* (35), *Ocean Sunfish* (55), *Ocelot* (1), *Octopus Oculifer* (5), *Odonus Niger* (50), *Office Chair Base Wheel* (44), *Off-Road Vehicle* (55), *Oil Tanker* (37), *Olive Sea Turtle* (34), *Olives* (11), *One-Shoulder Top* (3), *Onion* (4), *Opalescent Nudibranch* (35), *Opossum* (27), *Orange* (47), *Oriental Dollarbird* (25), *Oriental Small-Clawed Otter* (9), *Oriental Turtle-Dove* (54), *Ornate Octopus* (33),

*Oscillating Multi-Tool* (19), *Osmanthus Fragrans* (22), *Ostrich* (35), *Otter* (51), *Oven* (25), *Oversized Tote* (39), *Oxyeleotris Marmorata* (12), *Oyster Mushroom* (30).

## P (155 classes)

*Pacific Gull* (10), *Pacific Spiny Lumpsucker* (35), *Packing Cube* (38), *Pad* (46), *Paddle Go-Round* (10), *Paddle Wheeler Ride* (5), *Paddleball* (30), *Paddlefish* (33), *Paddy* (2), *Paint Sprayer* (3), *Painted Frogfish* (35), *Painter* (93), *Pajama Cardinalfish* (25), *Pajamas* (3), *Pallass Fish Eagle* (35), *Pallass Rosefinch* (40), *Panamanian Balboa* (74), *Panel Saw* (20), *Pannier Bag* (50), *Pants* (3), *Paper Airplane* (6), *Paper Clip* (6), *Paper Cutter* (15), *Paper Nautilus* (37), *Papillon* (3), *Parablennius Gattorugine* (26), *Parachanna Africana* (40), *Paracheilinus Bellae* (10), *Paracheilinus Cyaneus* (3), *Paracheilinus Hemitaeniatus* (4), *Paracheilinus Octotaenia* (19), *Paracheilinus Piscilineatus* (16), *Paracheilinus Rubricaudalis* (6), *Paraclinus Spectator* (1), *Paradise Fish* (49), *Pareques Acuminatus* (19), *Parkour Practitioners* (1), *Parodia Cactus* (11), *Passenger Aircraft* (65), *Pasta Server* (8), *Pata* (3), *Pca Pump* (10), *Pea* (6), *Peacock* (8), *Peacock Grouper* (14), *Peacock Gudgeon* (34), *Peanutt* (9), *Pear* (41), *Pecan* (3), *Pedal Karts* (3), *Peeler* (6), *Pen Needle* (3), *Pencil* (50), *Pencil Case* (27), *Pepper* (4), *Pepper Mill* (9), *Percnon Gibbesi* (11), *Persian Cat* (2), *Peruvian Apple Cactus* (10), *Peruvian Booby* (8), *Peruvian Night Monkey* (4), *Peruvian Sol* (5), *Peyote* (6), *Ph Meter* (99), *Philippine Tarsier* (3), *Phone* (49), *Phone Case* (117), *Phone Charger* (28), *Photochromic Lenses* (1), *Pianist* (35), *Picasso Triggerfish* (34), *Picc Line* (3), *Piccolo* (32), *Pickup Truck* (27), *Picnic Basket* (26), *Pied Tamarin* (3), *Piglet Squid* (20), *Pig-Tailed Macaque* (34), *Pikachu Nudibranch* (35), *Pillbug* (5), *Pilot Boat* (57), *Pinafore* (3), *Pine Marten* (10), *Pine Nut* (34), *Pink-Browed Rosefinch* (20), *Pipa* (35), *Pipe Stand* (5), *Pipe Vise* (18), *Pipe Wrench* (15), *Pistachio* (1), *Pizza Stone* (8), *Plaid Shirt* (3), *Plains Zebra* (9), *Plant Box* (26), *Plaster* (21), *Plate* (29), *Platform Shoes* (97), *Platy Fish* (35), *Platypus* (26), *Playground Swing Set* (30), *Playing Cards* (5), *Plug Strip* (59), *Plum Ak74* (3), *Plumb Bob* (8), *Plum-Headed Parakeet* (33), *Pneumatic Drill* (1), *Pod* (4), *Pokemon* (45), *Polarized Sunglasses* (2), *Police Baton Training* (16), *Polleni Grouper* (8), *Polo* (15), *Polynemus Paradiseus* (15), *Pomarine Skua* (10), *Pomegranate* (89), *Pomeranian* (1), *Poodle* (20), *Popcorn Box* (35), *Poppies* (6), *Porpoise* (9), *Port Jackson Shark* (37), *Portable Buoyant Device* (88), *Portable Spotlight* (20), *Portly Spider Crab* (11), *Potato* (3), *Potato Masher* (8), *Potato Ricer* (7), *Pottery Figurine* (218), *Power Broom* (18), *Power Float* (3), *Power Screed* (20), *Ppk* (37), *Ppq* (30), *Prefilled Syringe* (3), *Pressure Cooker* (5), *Pressure Gauge* (35), *Prince Of Persia* (4), *Printed Dress* (3), *Printer Paper* (30), *Printing Press Plate Cylinder* (60), *Proboscis Monkey* (30), *Protective Clothing* (4), *Protist* (31), *Prototype Racing Car* (48), *Protractor* (95), *Pudao* (10), *Pufferfish* (32), *Pull-Up Bars* (9), *Pumpkin* (4), *Pumpkin Seedt* (9), *Putty Knife* (18), *Pygmy Cormorant* (10), *Pygmy Gourami* (36), *Pygmy Hog* (28), *Pygmy Squid* (16).

## Q (4 classes)

*Qanun* (35), *Qipao Cheongsam* (4), *Quail Egg* (26), *Quinoa* (6).

## R (107 classes)

*Raccoon* (34), *Racing Car* (34), *Radish* (2), *Ragdoll Cat* (3), *Railroad Crossing Ahead Sign* (59), *Rain Boots* (9), *Rainbow Shark* (34), *Rainbow Shirt* (2), *Rapier* (169), *Razor* (29), *Razorbill* (3), *Rc Boat* (35), *Reading Glasses* (11), *Rebab* (33), *Reciprocating Saw* (20), *Reconstitution Syringe* (3), *Record Player Turntable* (42), *Red Arrow Raw 15* (3), *Red Bean* (6), *Red Damselfly* (19), *Red Deer* (25), *Red Firefish* (35), *Red Large-Toothed Snake* (3), *Red River Hog* (53), *Red Stumpnose* (13), *Red Velvetfish* (12), *Red Fox* (30), *Red-Billed Blue Magpie* (91), *Red-Billed Leiothrix* (15), *Red-Browed Finch* (65), *Red-Crowned Crane* (40), *Red-Faced Spider Monkey* (4), *Redflank Bloodfin* (3), *Red-Flanked Bluetail* (35), *Red-Lipped Batfish* (35), *Red-Tailed Black Shark* (10), *Red-Tailed Pipe Snake* (3), *Red-Wattled Lapwing* (32), *Red-Winged Blackbird* (34), *Reeves S Muntjac* (31), *Reindeer* (61), *Remote Control* (90), *Remote Control Airplane* (196), *Reporter* (13), *Rescue Basket Stretcher* (11), *Rescue Board* (52), *Rescue Breathing Equipment* (1), *Rescue Buoy* (3), *Rescue Buoy With Lights* (5), *Rescue Sled* (6), *Rescue Snap Hook* (80), *Rescue Strobe* (16), *Rescue Throw Line* (20), *Rescue Tube* (12), *Rescue Whistle* (11), *Rhinoceros* (35), *Rhinoceros Auklet* (5), *Rhythmic Gymnastics With Clubs* (30), *Ribbon Seal* (48), *Richards Pipit* (49), *Rickshaw* (45), *Ring* (107), *Ring-Tailed Lemur* (31), *River Tern* (10), *Riverboat* (20), *Robe* (3), *Robot* (22), *Robot Dog* (51), *Rock Shag* (4), *Rocket* (140), *Rocket Plane* (15), *Rocking Horse* (60), *Roe Deer* (24), *Roller Coaster* (43), *Roller Hockey* (4), *Roller Hockey Puck* (50), *Roller Skate* (8), *Roller Skating* (11), *Roller Speed Skating* (30), *Rolling Barrels* (3), *Rolling Pin* (10), *Rolling Tool Tote* (3), *Rope Bridge* (1), *Roseline Shark* (22), *Rotary Cutter Blade* (70), *Rotary Hammer* (18), *Rotating Climber* (5), *Rough Road Sign* (5), *Round Glasses* (3), *Roundabout Sign* (12), *Rower* (25), *Rowing* (48), *Royal Tern* (9), *Rpk* (3), *Rpk74* (3), *Rpk74M* (3), *Ruan* (32), *Ruby-Throated Hummingbird* (35), *Ruffe Fish* (6), *Ruger Gp100* (37), *Ruger P90* (32), *Ruger Precision Rifle* (3), *Ruger Sr9C* (60), *Russian Blue Cat* (5), *Rv* (31), *Ryegrass* (5).

## S (233 classes)

*Safety Glasses* (2), *Safety Needle* (2), *Saiga Antelope* (8), *Sailboat* (31), *Sailfish* (8), *Sailing Dinghy* (40), *Salad Dressing Bottle* (6), *Salvia Splendens* (33), *Samoyed* (13), *Sand And Water Table* (6), *Sand Diggers* (10), *Sandals* (38), *Sandbox Toys* (3), *Sanxian* (19), *Sar21* (1), *Sarangi* (26), *Sarrusophone* (1), *Saucepan* (10), *Savage Impulse* (3), *Sawshark* (17), *Saxophone* (34), *Scalpel* (17), *Scar* (3), *Scarf* (94), *Scarlet Finch* (41), *Sccy Cpx-1* (10), *Schmidt-Rubin M1889* (2), *School Bus* (32), *School Zone Sign* (30), *School Zone Speed Limit Sign* (39), *Scissor Lift* (13), *Scolopsis Bilineata* (12), *Scooter* (8), *Scottish Fold Cat* (4), *Screw Gun* (9), *Scrippss Murrelet* (1), *Scrubber* (16), *Sea Angel* (22), *Seahorse* (34), *Security Guard* (13), *Semi-Rimless Glasses* (2), *Sepak Raga* (10), *Sepak Takraw* (30), *Sergeant Major Fish* (44), *Severum Cichlid* (35), *Sewer Cleaning Truck* (27), *Sewing Thread* (50), *Shadow Play Area* (3), *Shakuhachi* (24), *Shaolin Stick Fighting* (30), *Sharp Turn Sign* (1), *Shawl* (7), *Sheep* (37), *Shell Game Cup* (12), *Shiba Inu* (12), *Shield Bug* (5), *Shillelagh* (34), *Ship Is Wheel* (39), *Shirt* (4), *Shoe Box* (32), *Shoes* (97), *Shogi* (5), *Shopping Cart Wheel* (38), *Shorts* (4), *Shot Put* (4), *Shoulder Drop-Off Sign* (1), *Shrew* (12), *Shuttlecock* (30), *Siamese Cat* (4), *Siberian Husky* (15), *Siberian Rubythroat* (49), *Siberian Thrush* (4), *Siberian Weasel* (34), *Sig P220S* (3), *Sig P226* (11), *Sig Sauer M400 Tread* (3), *Sig-Sauer P230* (7), *Sika Deer* (30), *Silver Dollar Fish* (12), *Silver Torch Cactus* (4), *Silvery Marmoset* (3), *Singer* (64), *Sitar* (35), *Site Light* (3), *Skate* (22), *Skateboard* (6), *Skateboard Ramp* (18), *Skateboard Wheel* (48), *Skateboarder* (69), *Ski Pole* (98), *Skillet* (9), *Skirt* (4), *Skydiving Aircraft* (26), *Sleeping Bag* (12), *Sling Bag* (1), *Sling Seat Swing* (60), *Slipform Paver* (8), *Slipper Lobster* (25), *Slippers* (4), *Slippery Road Sign* (2), *Slot Car* (56), *Slot Machine Wheel* (20), *Sloth Bear* (34), *Slow Children Playing Sign* (14), *Slow Loris* (10), *Slow Signal Sign* (7), *Slr Camera Len* (45), *Snailfish* (19), *Sneaker* (30), *Snooker* (35), *Snow Leopard* (34), *Snowboard* (29), *Snowmobile* (34), *Snowmobile Track Wheel* (58), *Snowplow* (35), *Snowshoe Hare* (35), *Snowy Owl* (31), *Soap Box* (29), *Soccer Player* (11), *Socks* (4), *Solar-Powered Boat* (31), *Soldier* (59), *Sooty Gull* (8), *Sound Velocity Meter* (11), *Sousaphone* (29), *South African Rand* (85), *South Korean Won* (36), *South Polar Skua* (1), *Soy Milk* (22), *Soy Sauce Bottle* (9), *Soybean* (4), *Spanish Dancer* (35), *Spatula* (9), *Spear Thrower* (8), *Spectacled Caiman* (34), *Spectacled Guillemot* (5), *Spectrophotometer* (28), *Speed Bump Sign* (5), *Spell* (55), *Sphygmomanometer* (25), *Sphynx Cat* (4), *Spider* (5), *Spin Zone* (9), *Spinal Needle* (3), *Spinning Platforms* (9), *Spinning Top* (3), *Spiral Climber* (2), *Spiral Spinners* (1), *Spiral Vegetable Slicer* (7), *Spirit Level* (44), *Splash Pads* (2), *Sponge Crab* (43), *Spoon* (34), *Sports Car* (30), *Spotted Hatchetfish* (35), *Spotted Knifejaw* (25), *Spotted-Tail Quoll* (32), *Spray Paint Wheel* (70), *Spring Rider* (15), *Springfield Armory Waypoint* (3), *Springfield Xdm-10* (20), *Square Dancer* (29), *Squash* (48), *Squash Racket* (30), *Squirrel* (6), *Squirrelfish* (27), *Sr-25* (3), *Ss2* (1), *Staple Gun* (10), *Stapler* (28), *Starcraft* (33), *State Highway Sign* (5), *Statues Of Buddha And Deities* (3), *Steamer* (23), *Steamers* (6), *Steel Drums* (31), *Steel Toe Boots* (34), *Steel Trowel* (2), *Steep Hill Sign* (5), *Steering Wheel* (48), *Stejneger S Pit Viper* (3), *Stethoscope* (12), *Stick Fighting Arts* (30), *Sticky Note* (1), *Stingrays* (24), *Stock Pot* (10), *Stopcock* (5), *Strainer* (9), *Strawberry* (39), *Street Dancer* (69), *Street Pedestrian* (66), *Street Sweeper* (46), *Striped Boarfish* (35), *Striped Burrfish* (3), *Striped Dolphin* (23), *Striped Hyena* (31), *Striped Pyjama Squid* (30), *Striped Shirt* (3), *Striped Snakehead* (9), *Stroller Wheel* (55), *Suit* (3), *Sun Bear* (74), *Sun Hat* (51), *Sunflower* (48), *Sunflower Seedt* (5), *Sunglasses* (6), *Super Mario* (35), *Superagui Lion Tamarin* (3), *Surfboard* (27), *Surfer* (32), *Swab* (15), *Swallows* (46), *Swallow-Tailed Gull* (4), *Sweater* (3), *Swedish Krona* (4), *Sweeper Truck* (90), *Sweeping Robot* (55), *Sweet Potato* (4), *Swimwear* (4), *Swing Gliders* (1), *Swing The Ball* (24), *Swiss Franc* (20), *Sword And Fairy* (34), *Swordfish* (8), *Swordtail Fish* (34), *Synodontis Grandiops* (21),

*Synodontis Multipunctatus* (47), *Synthesizer* (30), *Syrian Hamster* (33), *Syringe* (38), *Syringe Cap* (2), *Syringe Filter* (2), *Syringe Pump* (2).

## T (114 classes)

*T Intersection Sign* (6), *Table Tennis* (1), *Table Tennis Bats* (19), *Table Tennis Player* (39), *Tac-50* (3), *Taiko Drum* (29), *Takin* (35), *Tamarind Fruit* (4), *Tango Dancer* (30), *Tank* (55), *Tanto* (13), *Tape Measure* (34), *Tar* (7), *Tar-21* (2), *Tasmanian Devil* (31), *T-Bevel* (3), *Tea* (21), *Tea Tree Mushroom* (2), *Teeter-Totter* (20), *Tekko Kagi* (11), *Telescope* (11), *Telescopic Handler* (3), *Television* (28), *Tem* (5), *Temmincks Courser* (9), *Temporary Road Closed Sign* (2), *Tennis* (30), *Tennis Racket* (30), *Tensile Testing Machine* (4), *Terapon Jarbua* (42), *Testudinidae* (29), *Texas Prickly Pear Cactus* (5), *Thalasseus Sandvicensis* (8), *The Witcher* (26), *Theorbo* (16), *Thermometer* (14), *Thickness Gauge* (100), *Thimble Cactus* (24), *Thorny Devil* (30), *Thread Gauge* (33), *Three-Section Staff* (35), *Threestripe Gourami* (20), *Thresher Shark* (34), *Throw Bag* (73), *Throwing Axe* (34), *Throwing Knife Demonstrations* (29), *Tiara* (105), *Tibetan Mastiff* (19), *Tibetan Antelope* (35), *Tiger* (27), *Tiger Barb* (11), *Tiger Hook Swords* (33), *Tiger Shark* (35), *Tile Spacers* (2), *Tilting Platforms* (1), *Timer* (1), *Timpani* (28), *Tin Whistle* (29), *Tire Swing* (36), *Tissue Box* (34), *Toddler Play Area* (2), *Toilet Brush* (10), *Toll Booth Sign* (1), *Tom Thumb Cactus* (3), *Tomahawk* (31), *Tomato* (2), *Tomato Slicer* (4), *Tongs* (8), *Toothpaste Bottle* (4), *Torch Cactus* (33), *Torque Wrench* (44), *Tortoise Beetle* (5), *Tote Bag* (33), *Touchscreen Gloves* (4), *Tourist Guide* (9), *Toy Box* (5), *Toy Car Wheel* (31), *Toy Piano* (29), *Track And Field Athlete* (30), *Trackless Train* (3), *Tracksuit* (4), *Traditional Chinese Sword Dance* (24), *Traditions Pursuit Xt* (3), *Train* (31), *Tray* (3), *Treefish* (10), *Trench Club* (27), *Trench Coat* (3), *Trencher* (3), *Triangle* (35), *Triangle Board* (5), *Trichocereus* (10), *Tripod Stand* (20), *Trocar Needle* (2), *Trolley* (30), *Trolley Bag* (41), *Trombone* (48), *Truck Racing Truck* (48), *Trumpet* (51), *T-Shirt* (3), *T-Shirt Dress* (2), *Tuba* (34), *Tufted Capuchin* (3), *Tufted Deer* (35), *Tufted Puffin* (4), *Tugboat Ride* (3), *Tumble Drums* (7), *Tumble Tracks* (6), *Tuna Crab* (24), *Tunnel Maze* (2), *Turkish Lira* (2), *Turks Cap Cactus* (4), *Turtleneck* (3), *Tyrannosaurus Rex* (28).

## U (11 classes)

*Uav* (48), *Ukulele* (49), *Umbrella* (31), *Underwear* (8), *Uniform* (4), *United Arab Emirates Dirham* (2), *Upside-Down Jellyfish* (35), *Us Dollars* (3), *Utility Knife* (10), *Uv Sterilizer Vacuum* (26), *Uv-Vis* (3).

## V (18 classes)

*Vacuum Cleaner* (40), *Vacuum Cleaner Brush Roller* (99), *Vacuum Cleaner Crevice Tool Wheel* (18), *Vampire Squid* (25), *Vanilla Bean* (7), *Vernier Caliper* (107), *Vest* (8), *Vial Adapter* (1), *Vibration Meter* (34), *Vietnamese Leaf Monkey* (8), *Vintage Aircraft* (35), *Vintage Car* (40), *Violinist* (12), *Virtual Reality Headset* (3), *Virtual Reality Rides* (4), *Viscometer* (50), *Volleyball* (30), *Volleyball Player* (49).

## W (60 classes)

*Waffle Wheel* (110), *Wagon Wheel* (1), *Waiter* (35), *Walkie Talkies* (31), *Wall Decoration Painting* (12), *Wall Scanner* (20), *Wallaby* (34), *Wallball* (30), *Wallet* (30), *Walnut* (3), *Walrus* (28), *Waltz Dancer* (23), *Waspfish* (43), *Water Bottle* (2), *Water Filter Bottle* (29), *Water Gun* (28), *Water Painting Wall* (3), *Water Play Table* (3), *Water Polo* (18), *Water Shooters* (10), *Water Walking Balls* (10), *Water Wheel* (34), *Watering Truck* (35), *Wave Slide* (1), *Weddell Seal* (32), *Wedges* (83), *Weekender* (47), *Weever Fish* (28), *Weigthlifter* (34), *Western Pygmy Marmoset* (2), *Wetmorella Nigropinnata* (16), *Whale Shark* (14), *Wheat* (3), *Wheelchair* (209), *Wheelchair Wheel* (50), *Whirling Whirlpools* (4), *Whisk* (8), *White-Cheeked Spider Monkey* (5), *White-Faced Saki* (3), *White-Fronted Tern* (9), *White-Headed Marmoset* (3), *White-Lipped Deer* (15), *White-Lipped Tamarin* (3), *White-Naped Crane* (43), *White-Nosed Saki* (2), *White-Tailed Rubythroat* (35), *White-Throated Rock Thrush* (10), *Whooper Swan* (50), *Wide-Brimmed Hat* (3), *Wieds Marmoset* (3), *Wild Boar* (33), *Wind Speed Gauge* (4), *Wine Glass* (49), *Wire Stripper* (3), *Wok* (40), *Wolverine* (25), *Workwear* (7), *Wristwatch Wheel* (9), *Writing Brush* (29), *Wurlitzer Piano* (32).

## X (3 classes)

*Xema Sabini* (4), *Xm109* (1), *Xylophone* (32).

## Y (17 classes)

*Yangqin* (31), *Yellow Crested Weedfish* (14), *Yellow Peach* (23), *Yellow Warbler* (34), *Yellow-Bellied Tit* (34), *Yellow-Billed Magpie* (33), *Yellow-Cheeked Tit* (45), *Yellowfin Flasher Wrasse* (8), *Yellowmargin Triggerfish* (28), *Yellow-Rumped Warbler* (31), *Yellowtail Damselfish* (35), *Yellow-Throated Marten* (35), *Yellow-Throated Warbler* (5), *Yoga Mat Bag* (42), *Yoga Wheel* (34), *Yo-Yo* (30), *Y-Site Injection Port* (1).

## Z (9 classes)

*Zebra* (33), *Zebra Danio* (34), *Zebra Finch* (45), *Zebra Loach* (47), *Zebra Pleco* (45), *Zebra Shark* (40), *Zhanmadao* (3), *Zhongshan Suit* (3), *Zither* (32).

## A.2 Statistic of Video Length

In Fig. 11, we show distribution of video length on VastTrack. Please notice, VastTrack has an average video length of 83 frames, and mainly focuses on short-term tracking. Despite this, VastTrack can still be applied to train long-term temporal trackers, as evidenced by the effectiveness of short-term videos for developing robust trackers in both short- and long-term scenarios [27].

## A.3 Summary of Evaluated Trackers

For better understanding of the evaluated trackers, we provide a summary in Tab. 5. All these trackers are evaluated as they as without modifications. It is worth noting that, the work of DropMAE is marked with "TP" is because it employs temporal information for pre-training.

## A.4 Full Results of Attribute-based Evaluation

We present evaluation for all trackers under ten attributes. Specifically, Fig. 12, Fig. 13, and Fig. 14 respectively show the attribute-based evaluations using PRE, NPRE, and SUC. From these evaluation

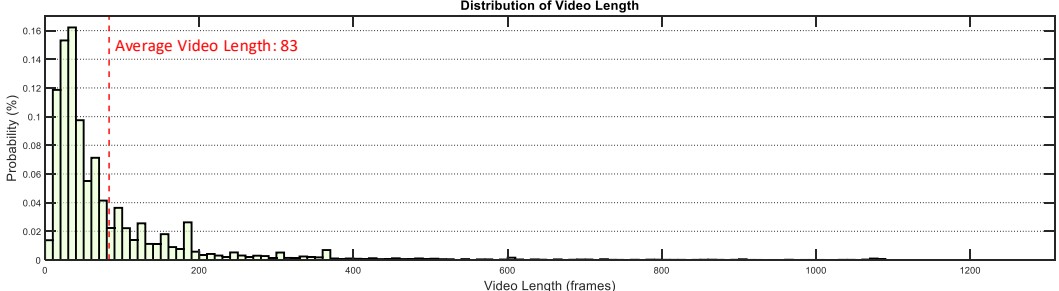

Figure 11: Distribution of sequence length on VastTrack. The average video length of VastTrack is 83 frames (∼14 seconds with frame rate 6 *fps*). Please note that, VastTrack is focused on short-term tracking, but could also be used for training long-term temporal trackers as discussed in the main text.

Table 5: Summary of the evaluated tracking algorithms. "CNN": CNN-based; "CNN-T": CNN-Transformer-based, "Trans.": Transformer-based. TP: "✓" for trackers leveraging temporal information, and "✗" for trackers using only the information from initial frame for tracking.

| Tracker | Where | Backbone | Type | TP |
|---|---|---|---|---|
| SiamFC [1] | ECCVW'16 | AlexNet | CNN | ✗ |
| ATOM [11] | CVPR'19 | ResNet-18 | CNN | ✓ |
| SiamRPN++ [33] | CVPR'19 | ResNet-50 | CNN | ✗ |
| DiMP [2] | ICCV'19 | ResNet-50 | CNN | ✓ |
| SiamBAN [8] | CVPR'20 | ResNet-50 | CNN | ✗ |
| SiamCAR [23] | CVPR'20 | ResNet-50 | CNN | ✗ |
| PrDiMP [12] | CVPR'20 | ResNet-50 | CNN | ✓ |
| Ocean [60] | ECCV'20 | ResNet-50 | CNN | ✓ |
| STMTrack [20] | CVPR'21 | GoogLeNet | CNN | ✓ |
| TrSiam [49] | CVPR'21 | ResNet-50 | CNN-T | ✓ |
| TransT [7] | CVPR'21 | ResNet-50 | CNN-T | ✗ |
| STARK [56] | ICCV'21 | ResNet-101 | CNN-T | ✓ |
| AutoMatch [59] | ICCV'21 | ResNet-50 | CNN | ✗ |
| ToMP [40] | CVPR'22 | ResNet-101 | CNN-T | ✓ |
| MixFormer (L) [9] | CVPR'22 | CVT24-W | Trans. | ✓ |
| OSTrack (384) [57] | ECCV'22 | ViT-Base | Trans. | ✗ |
| RTS [44] | ECCV'22 | ResNet-50 | CNN | ✓ |
| SimTrack (224) [5] | ECCV'22 | ViT-Large | Trans. | ✗ |
| SwinTrack (224) [38] | NeurIPS'22 | SwinT | Trans. | ✓ |
| SeqTrack (L384) [6] | CVPR'23 | ViT-Large | Trans. | ✓ |
| ARTrack (256) [51] | CVPR'23 | ViT-Large | Trans. | ✓ |
| DropMAE [52] | CVPR'23 | ViT | Trans. | ✓ |
| GRM (256) [22] | CVPR'23 | ViT-Base | Trans. | ✗ |
| ROMTrack (384) [4] | ICCV'23 | ViT-Base | Trans. | ✓ |
| MixFormerV2 (B) [10] | NeurIPS'23 | ViT-Base | Trans. | ✓ |

results, we can observe that, existing state-of-the-art trackers heavily suffer from various challenges in the videos. To achieve general tracking, more efforts are needed to improve their robustness.

## A.5 Maintenance, Ethical Issue, and Responsible Usage of VastTrack

**Maintenance.** Our VastTrack is hosted on Github (https://github.com/HengLan/VastTrack). This allows us to conveniently check feedback from the community and to improve VastTrack via necessary maintenance and updates by research groups of senior authors of VastTrack. Besides, we'll try our best to continue assessing future trackers to offer up-to-date evaluation and comparison on VastTrack. The ultimate goal is to offer a long-term and stable platform for the tracking community.

**Ethical Considerations Related to Videos.** We avoid using private videos and all videos are collected under the Creative Commons license for research only. However, we understand the license might change in future. Once any notification regarding this is received, we'll take action to handle it.

**Guidelines for the Responsible Use of VastTrack.** The development of VastTrack aims to facilitate the research and application of tracking. It can be used for *research purpose only*. Please note that, due to inevitable bias during data collection, there may exist geographic and demographic imbalance.

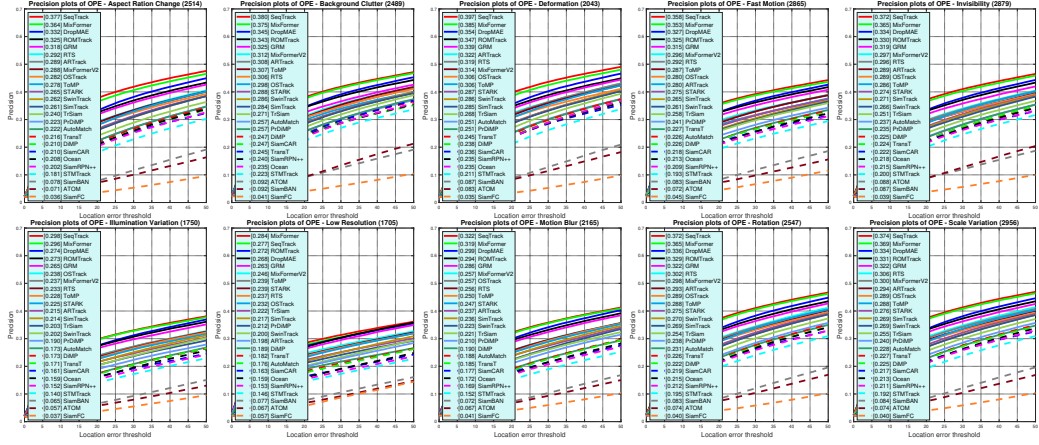

Figure 12: Performance of trackers on ten attributes, including ARC, BC, DEF, FM, INV, IV, LR, MB, ROT, and SV, using precision (PRE).

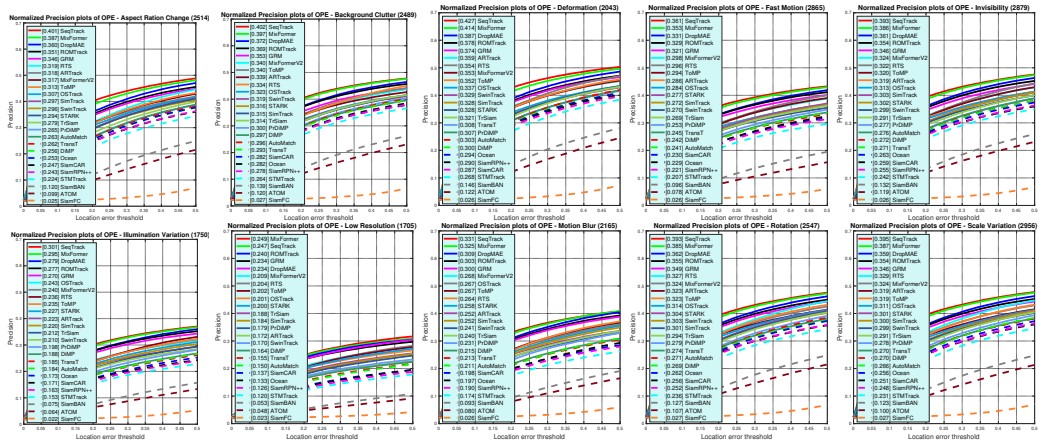

Figure 13: Performance of trackers on ten attributes of ARC, BC, DEF, FM, INV, IV, LR, MB, ROT, and SV using normalized precision (NPRE).

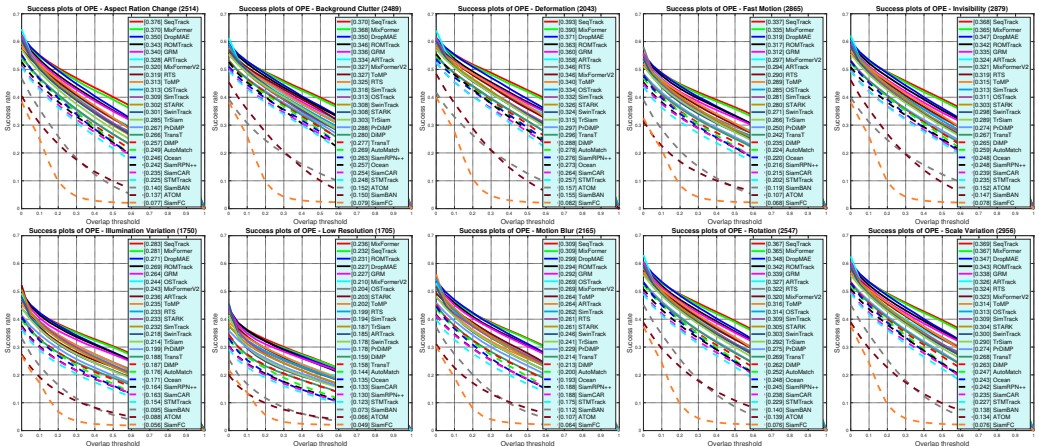

Figure 14: Performance of trackers on ten attributes, including ARC, BC, DEF, FM, INV, IV, LR, MB, ROT, and SV, using success (SUC).

