# OpenReview forum: "VastTrack: Vast Category Visual Object Tracking"
_NeurIPS.cc/2024/Datasets_and_Benchmarks_Track — NeurIPS 2024 Track Datasets and Benchmarks Poster_

### Official Review · Reviewer_LM87 · 2024-07-22
**Review of VastTrack**

**Rating:** 7
**Confidence:** 5
**Correctness:** Basically, the claims made in the sub…
**Clarity:** The written is easy to follow.

**Review:**

The contributions of this work are significant. But I have some questions about this work.

- As we all know, single object tracking is a class-independent task, which aims to learn the similarity with a target given at the first frame. So, whether it is more important to train a tracker for non-specific classes, which can work on a “UNSEE” new class, but not to “SEE” more classes in the training dataset.

- As claimed by the authors, VastTrack is focused on short-term tracking by offering abundant object classes and sequences. However, long-term tracking seems to be more challenging and practical. Although the training on short videos is helpful, at least the testing on long videos is more required.

- As shown in Table, the retraining on VastTrack is not very significant. On testing of VastTrack itself with various classes, even without the training, the performance is acceptable. This demonstrates that the tracker on previous classes can handle the vast unseen classes. Also, after training on VastTrack with many more classes, the performance improvement is not very large.

**Strengths:**

+ VastTrack consists of videos from 2,115 classes, which largely surpasses category numbers in popular benchmarks such as GOT-10k and LaSOT.

+ VastTrack offers 50,610 video sequences with 4.2 million frames, which makes it so far the largest and the most diverse tracking dataset in terms of the number of videos and targets compared to existing datasets.

+ VastTrack offers both standard bounding box annotations and rich linguistic specifications for the sequences.

+ The authors claim that, to ensure precise annotations, each video in VastTrack is manually labeled with multi-round refinements.

[+] I like the distinctive symbol used in the contribution.

**Additional Feedback:**

N/A

**Documentation:**

Basically sufficient.

**Ethics:**

I have no ethical concerns with the submission.

**Limitations:**

- As we all know, single object tracking is a class-independent task, which aims to learn the similarity with a target given at the first frame. So, whether it is more important to train a tracker for non-specific classes, which can work on a “UNSEE” new class, but not to “SEE” more classes in the training dataset.

- As claimed by the authors, VastTrack is focused on short-term tracking by offering abundant object classes and sequences. However, long-term tracking seems to be more challenging and practical. Although the training on short videos is helpful, at least the testing on long videos is more required.

- As shown in Table, the retraining on VastTrack is not very significant. On testing of VastTrack itself with various classes, even without the training, the performance is acceptable. This demonstrates that the tracker on previous classes can handle the vast unseen classes. Also, after training on VastTrack with many more classes, the performance improvement is not very large.

**Opportunities For Improvement:**

- Besides Table 4, more experiments are suggested to show the advantage of VastTrack. For example, tracking on VastTrack can improve the performance of other datasets. Also, the experiments showing the challenges of the testing set of VastTrack are helpful.

- The Transformer-based method has reached 2023. The CNN-based method and CNN-Transformer-based method have both reached 2022, whether there have been no such methods in 2023? Are there more recent SOT works on these three categories of methods in 2024?

- The category name at the very edge of Fig1 in the supplementary material is not very clear.

**Relation To Prior Work:**

This work has discussed how this work differs from previous contributions.

**Summary And Contributions:**

In this paper, the authors build a new large benchmark, named VastTrack, which aims to facilitate the development of visual tracking by encompassing more classes and videos. It covers targets from 2,115 categories and provides 50,610 videos with 4.2 million frames. VastTrack also provides linguistic descriptions with more than 50K sentences for the videos. To understand the performance of existing trackers and to provide baselines for future comparison, this work also extensively evaluates 25 representative trackers.

---

> ### Author Rebuttal · Authors · 2024-08-16
>
> We thank the reviewer for helpful comments on our work and offer our responses below.
>
> >**Q1:** Question on whether it is more important to train a tracker for non-specific classes, which can work on a "UNSEE" new class, but not to "SEE" more classes in the training dataset.
>
> **A1:** Thanks for this insightful comment. We understand and agree that tracking is an instance-level task and needs to be trained for non-specific class. The purpose of VastTrack is to provide abundant object classes as well as various instances from these classes, which could increase the ability of trackers in handling unseen new classes, towards universal tracking. The possible reason behind this is, similar to human that see numerous classes and targets when developing tracking ability, abundant classes and instances in training may allow the tracker to learn more general representation, which can generalize to tracking targets even from unseen classes. Thus motivated, we introduce VastTrack and expect it to further facilitate the generality of tracking. We’ll clarify this in revision. Thanks!
>
> >**Q2:** As claimed by the authors, VastTrack is focused on short-term tracking by offering abundant object classes and sequences. However, long-term tracking seems to be more challenging and practical. Although the training on short videos is helpful, at least the testing on long videos is more required.
>
> **A2:** Thanks for this comment. We agree long-term tracking may be more practical. Our current priority is to advance short-term tracking by offering rich diversity and abundant videos (, which also benefits training of long-term tracking as mentioned). As suggested, in the future, we plan to design a new subset with around 1k videos for long-term tracking evaluation.
>
> We’ll clarify this in revision. Thanks!
>
> >**Q3:** The retraining on VastTrack is not very significant.
>
> **A3:** Thanks for this careful comment. The retraining experiment is conducted using same setting of the original tracker. Even without specific tuning, retraining improves SiamRPN++ with 1.7\% and OSTrack with 2.6\% performance gains on VastTrack test set. Please note, these numbers show obvious improvements in the tracking community. Moreover, if tuning the parameters, we believe the performance on VastTrack will be further increased in dealing with more classes.
>
> We'll clarify the above in revision. Thanks!
>
> >**Q4:** More experiments are suggested to show the advantage of VastTrack, like **(a)** tracking on VastTrack can improve the performance of other datasets and **(b)** the experiments showing the challenges of the testing set of VastTrack are helpful.
>
> **A4:** Thanks for these helpful comments.
>
> **(a)** We think the reviewer might mean "training on VastTrack" instead of "tracking on VastTrack". In this work, we conduct experiments by retraining SiamRPN++ and OSTrack on VastTrack and assessing them on the other dataset LaSOT. The SUC score of SiamRPN++ on LaSOT is increased from 0.496 to 0.528 with 3.2% gains, and the SUC score of OSTrack is improved from 0.711 to 0.722 with 1.1% gains (please note, OSTrack is already strong on LaSOT but still boosted using VastTrack). Moreover, we assess them on two more datasets TrackingNet and TNL2K. On TrackingNet, SiamRPN++ is improved from 0.733 to 0.762 with 2.9% gains in SUC, and OSTrack from 0.839 to 0.852 with 1.3% gains. On TNL2K, SiamRPN++ is boosted from 0.413 to 0.446 in SUC with 3.3% gains, and OSTrack from 0.559 to 0.579 with 2.0% gains. All these show the usefulness of VastTrack. Besides the adopted ones, we’ll add more datasets for experiments in revision.
>
> **(b)** Thanks and we agree with the reviewer. In this work, we show the performance of trackers under different challenges (e.g., deformation, occlusion, fast motion, etc) on the testing set in Fig. 4, 5, and 6 in the supplementary material (please kindly see the supplementary material). As suggested, we will include new visual analysis of the trackers under these different scenarios in revision for showing more detailed challenges in our test set.
>
> We'll add the above analysis and results in revision. Thanks again!
>
> >**Q5:** The Transformer-based method has reached 2023. The CNN-based method and CNN-Transformer-based method have both reached 2022, **(a)** whether there have been no such methods in 2023? **(b)** Are there more recent SOT works on these three categories of methods in 2024?
>
> **A5:** Thanks for this comment. Please see below:
>
> **(a)** First, we'd like to clarify that, since there are numerous tackers each year, we mainly select trackers from major vision or learning conferences as they represent the trend in tracking. To our knowledge, inspired by the excellence of some trackers like MixFormer (CVPR'22) and OSTrack (ECCV'22), since 2023, trackers from major conferences all adopt pure Transformer architecture for tracking (all Transformer-based), and there are no CNN-based or CNN-Transformer-based trackers.
>
> **(b)** In major conferences in 2024, there are only Transformer-based trackers. To make our evaluation more up to date, we assess two recent trackers AQATrack [1] and HIPTrack [2] (both in CVPR'24) on VastTrack given they release codes and models. AQATrack obtains SUC/PRE/NPRE scores of 0.376/0.371/0.403, and HIPTrack SUC/PRE/NPRE scores of 0.386/0.370/0.401.
>
> [1] Autoregressive Queries for Adaptive Tracking with Spatio-Temporal Transformers, CVPR, 2024.
>
> [2] HIPTrack: Visual Tracking with Historical Prompts, CVPR, 2024.
>
> We'll include the above clarification and results in revision. Thanks again!
>
> >**Q6:** The category name at the very edge of Fig1 in supplementary material is not very clear.
>
> **A6:** Thanks for pointing this out. The class names are small because of too many categories. To make the hierarchical organization of categories clearer, we'll supplement the figure with an extra excel file (on project webpage) which organizes the hierarchy of categories in text for more clear display.
>
> Again, thanks!

---

> > ### Comment · Reviewer_LM87 · 2024-08-25
> > **Comments after rebuttal**
> >
> > Thanks for the authors' careful response, which has addressed most of my concerns. Therefore I keep my positive score of 'accept'.
> >
> > I hope the authors can improve this manuscript in the final version as they claimed in this response.
> >
> > A remaining question is about the previous Q3, I still don't think those numbers (provided in the response) are obvious improvements in the tracking community, considering the vast cost of this new training dataset. These improvement margins can also be obtained by optimizing the method, which is more economical. So I suggest the authors further explore the potential of the proposed vast dataset.
> >
> > Thanks.

---

> > > ### Author Response · Authors · 2024-08-25
> > > **Response to reviewer's comments**
> > >
> > > Dear Reviewer,
> > >
> > > Thank you for providing your thoughtful comments and constructive feedback on our work. We sincerely appreciate your careful review. As suggested, we will make necessary improvements to the manuscript as claimed in the rebuttal as well as conduct more explorations on our platform.
> > >
> > > Once again, thank you.

---

### Official Review · Reviewer_Czds · 2024-07-25
**The proposed VastTrack dataset is a novel and well-constructed benchmark dataset for visual object tracking in the real-world**

**Rating:** 7
**Confidence:** 4
**Correctness:** Technically sound.

**Review:**

Current visual tracking benchmarks are limited in the number of object categories, but the development of general object tracking systems capable of handling the vast diversity of objects encountered in real-world scenarios.

The authors aim to address this gap by providing a more comprehensive benchmark, VastTrack that includes a significantly larger number of object categories and videos.

Pros:
1) Vast object category coverage
2) Large scale
3) Rich and diverse annotations including 5K linguistic description for vision-language tracking research fields
4) High-quality annotations
5) Comprehensive evaluation

Cons:
1) Short video length
2) Low frame rates

**Strengths:**

Strengths are same as the summary and contributions

**Additional Feedback:**

.

**Clarity:**

This paper is well-written.

The contributions and novelty are clear, compared to other benchmark dataset.

Figures and tables shows them well.
Experiments and discussion are very clear as well.

**Documentation:**

In this paper, a URL is available for dataset (train/test) download by OneDrive and Baidu Cloud Drive.

The supplementary material also provided details of the proposed dataset (object classes, distribution of video length, etc.).

**Ethics:**

.

**Limitations:**

The authors provided possible limitations including short video length issue and social impact, noting that this benchmark is released for more general tracking systems and the advancement of tracking in real-world applications.

**Opportunities For Improvement:**

This benchmark dataset is limited to short-term tracking.

The average video length of the proposed benchmark is 83 frames with 6 fps.

In particular, 6 fps is significantly different from typical video sequences, which can impact video analysis for visual tracking.

**Relation To Prior Work:**

It is clearly discussed in sections 1 and 2.

In addition, the authors showed the difficulties of their benchmark in section 4.3 through the comparison to other datasets (Tables 3 and 4)

**Summary And Contributions:**

Its contributions are well-summarized in abstract:
1) Vast object category: 2,115 object classes (Fig. 2)
2) Larger scale: 50,610 videos (Fig. 1 and Table 1)
3) Rich annotation: bounding box + linguistic descriptions (Fig. 3)

---

> ### Author Rebuttal · Authors · 2024-08-16
>
> We appreciate the reviewer for careful comments and provide our responses below.
>
> >**Q1:** This benchmark dataset is limited to short-term tracking. The average video length of the proposed benchmark is 83 frames with 6 fps.
>
> **A1:** Thanks for this comment. We understand the reviewer's concern. However, in this work, the proposed VastTrack, similar to the popular Got-10k, is focused on advancing the short-term tracking by providing abundant object classes and sequences for both training and testing (Line 186 to Line 188 in the paper). Despite this, our VastTrack, like Got-10k, can still be applied to train long-term temporal trackers, as evidenced by the effectiveness of short-term videos ([26] in paper) for developing robust trackers in both short- and long-term scenarios. We believe that the diversity and quantity of objects and videos might be more crucial for deep tracking, which motivates the proposal of VastTrack to some extent.
>
> We thank the reviewer again for this comment and will include the above clarification in revision.
>
> >**Q2:** In particular, 6 fps is significantly different from typical video sequences, which can impact video analysis for visual tracking.
>
> **A2:** Thanks for this careful comment. We understand that the frame rate of our VastTrack is less than that of traditional benchmarks, but this may not impact the training and evaluation of tracking much. Specifically, on the training side, since most of the current tracking frameworks adopt the training mechanism of frame sampling with an interval (e.g., 100 frames), our VastTrack can be used for training by choosing a suitable interval (probably less than the interval used for traditional benchmarks). On the evaluation side, according to the analysis conducted in OxUvA ([44] in the paper), labeling at different frame rate, even at 1fps, does not adversely affect the robustness of tracking evaluation.
>
> Again, we thank the reviewer for this helpful comment, and will include the above clarification in revision.

---

### Official Review · Reviewer_5VYv · 2024-07-25
**new vision-language tracking benchmark**

**Rating:** 7
**Confidence:** 5

**Review:**

Quality: The paper is of high quality, offering a thorough and detailed examination of the proposed benchmark, VastTrack. The methodology for constructing the dataset and conducting the experiments is rigorous and well thought out.

Clarity: The writing is clear and concise. The authors effectively communicate the goals, methodology, and findings of their work. The structure of the paper allows for easy comprehension, even for readers who may not be experts in the field of visual tracking.

Originality: The work presents a significant original contribution to the field. The creation of VastTrack with its extensive number of object categories and video sequences is a novel approach that addresses current limitations in visual tracking benchmarks.

Significance: The significance of this work cannot be overstated. VastTrack has the potential to drive substantial advancements in visual tracking research by providing a more generalized and challenging dataset for algorithm development and evaluation.

**Strengths:**

1. A large-scale tracking dataset is proposed;
2. Comprehensive benchmark is provided.

**Additional Feedback:**

none.

**Clarity:**

Yes, the paper is well-written. The authors have effectively communicated their research objectives, the methodology employed, and the significance of their findings.

**Correctness:**

To ensure the correctness of the claims, the authors should provide additional details and comparisons to support their statements about the size and diversity of VastTrack. To support the reproducibility of the evaluation, the authors should provide more details on the selection criteria for the tracking algorithms, the hybrid evaluation protocol, and the computational resources used for the experiments.

**Documentation:**

For the dataset VastTrack presented in the paper, the authors have provided a reasonable level of detail regarding data collection and organization. However, there are areas where additional information would be beneficial to ensure thorough documentation.

Data Collection and Organization:
Sufficiency of Detail: The paper describes the process of collecting videos from various sources and the criteria for selecting the 2,115 object categories. However, the specifics of the data collection process, such as the tools and methods used for video acquisition, could be more detailed.
Annotation Process: The paper mentions the inclusion of rich annotations, including linguistic specifications, but the details of the annotation process, the guidelines followed by annotators, and the quality control measures, are not fully explained.
Availability and Maintenance:
Access to the Dataset: The paper should include a URL or other access information for reviewers to access the dataset. This is crucial for the review process to verify the claims made about the dataset’s content and quality.
Maintenance Plan: While the paper discusses the creation of the dataset, there is no clear mention of a plan for maintaining and updating VastTrack in the future, which is essential for its long-term utility.
Ethical and Responsible Use:
Ethical Considerations: The paper should address ethical considerations related to data collection, such as consent for using videos that may contain personal information or the potential for privacy breaches.
Responsible Use: The authors should provide guidelines or a code of conduct for the responsible use of the dataset, including how to handle potential biases in the data and the importance of fair and unbiased algorithm development.

**Ethics:**

I have no ethical concerns with the submission.

**Limitations:**

Have the authors adequately addressed the limitations and potential negative societal impact of their work? If not, please include constructive suggestions for improvement. In general, authors should be rewarded rather than punished for being upfront about the limitations of their work and any potential negative societal impact. You are encouraged to think through whether any critical points are missing and provide these as feedback for the authors.

figure 1 missing the TNL2K for the comparison; also, the TNL2K needs to be introduced into Table 3: Comparison to other datasets.
4.4 Retraining Experiments with VastTrack, the authors adopt two representative trackers for re-training. More recent language-based trackers are suggested to be evaluated.

**Opportunities For Improvement:**

1. figure 1 missing the TNL2K for the comparison; also, the TNL2K needs to be introduced into Table 3: Comparison to other datasets.
2. 4.4 Retraining Experiments with VastTrack, the authors adopt two representative trackers for re-training. More recent language-based trackers are suggested to be evaluated.

**Relation To Prior Work:**

The paper does provide a discussion on how the work differs from previous contributions, but there is room for improvement in terms of clarity and depth of this comparison.

Strengths in Relation to Prior Work:
Introduction of a New Benchmark: The authors clearly state the need for a new benchmark, VastTrack, by highlighting the limitations of existing datasets, such as the restricted number of object categories and the lack of diversity in scenarios.
Dataset Scale and Diversity: The paper emphasizes the scale and diversity of VastTrack, with 2,115 object categories and 50,610 video sequences, which is a significant advancement over previous benchmarks.
Rich Annotations: The inclusion of rich annotations, including linguistic specifications, is identified as a novel aspect of VastTrack, setting it apart from other datasets that typically only provide bounding box annotations.
Benchmarking Results: The paper presents a comprehensive benchmarking of various tracking algorithms on VastTrack and compares the results with those obtained on other datasets, demonstrating the challenges posed by the new benchmark.

**Summary And Contributions:**

The authors present a new large-scale benchmark called VastTrack for visual object tracking. The benchmark is designed to address the limitations of existing datasets by offering a vast number of object categories and a large number of video sequences. The dataset aims to facilitate the development of more generalized and universal tracking algorithms that can handle various objects and scenarios. The contributions of the paper include the introduction of VastTrack with rich and precise annotations, an extensive evaluation of 25 recent tracking algorithms on this new benchmark, and an analysis of the performance of these algorithms, which highlights the challenges in achieving universal tracking.

Contributions:
1. The authors introduce a new benchmark, VastTrack, which covers 2,115 object categories, significantly expanding the diversity of objects for tracking compared to existing benchmarks.
2. VastTrack provides a large scale of 50,610 videos, which is expected to benefit the development of more powerful deep learning-based trackers.
3. The dataset includes rich annotations, enabling the exploration of both vision-only and vision-language tracking approaches.
4. The authors conduct an evaluation of 25 trackers on VastTrack to understand its challenges and provide baselines for future comparisons.

---

> ### Author Rebuttal · Authors · 2024-08-16
>
> We thank the reviewer for careful comments and provide our responses below.
>
> >**Q1:** TNL2K is missing in Figure 1, and TNL2K needs to be introduced in Table 3 for comparison.
>
> **A1:** Thanks for your comments. We'll add TNL2K (cited as [48] in paper) in Fig. 1. Besides, as suggested, we have assessed the trackers in Tab. 3 on TNL2K and compare with VastTrack, as shown in Tab. A. From Tab. A, we can see VastTrack is more challenging. We'll add the comparison from Tab. A to Tab. 3 in revision. Thanks again for this comment!
>
> Tab. A: Comparison between TAN2K and VastTrack using SUC score.
> | Tracker | TNL2K | VastTrack |
> |--------------|-------|-----------|
> | SeqTrack | 0.578 | 0.396  |
> | MixFormer | 0.533 | 0.395  |
> | DropMAE | 0.569 | 0.375  |
> | ROMTrack | 0.604 | 0.370  |
> | GRM | 0.611 | 0.363 |
> | ARTrack | 0.575 | 0.356 |
> | RTS | 0.599 | 0.355 |
> | MixFormerV2 | 0.506 | 0.352 |
> | ToMP | 0.584 | 0.349 |
> | SimTrack | 0.556 | 0.344 |
> | OSTrack | 0.559 | 0.336 |
> | STARK | 0.525 | 0.334 |
> | SwinTrack  | 0.559 | 0.330 |
> | TrSiam | 0.523 | 0.326 |
> | PrDiMP | 0.470 | 0.310 |
>
> >**Q2:** More recent language-based trackers in retraining experiment.
>
> **A2:** Thanks for this helpful comment. As suggested, we plan to conduct experiments on the most recent language-based trackers (Context-Aware Integration of Language and Visual References for Natural Language Tracking, CVPR'24 and OneTracker: Unifying Visual Object Tracking with Foundation Models and Efficient Tuning, CVPR'24). However, since their implementations are not accessible at this moment, we’ll include them in revision once their implementations are available. Again, thanks!
>
> >**Q3:** More details on **(a)** the selection criteria for trackers, **(b)** hybrid evaluation protocol, and (c) computational resources for experiments for reproducible evaluation.
>
> **A3:** Thanks for these comments. Please see below:
>
> **(a) Selection criteria for trackers:** The selection criteria for trackers is to use impactful generic trackers from recent years. Please note, SiamFC is an exception. Despite being proposed long time ago, it’s chosen for evaluation as it’s a seminal work for existing Siamese framework. Based on this criterion, we select representative trackers from different types including CNN-based, CNN-Transformer-based, and Transformer-based. If the reviewer has suggestions to add specific trackers, we’ll include them in revision. Thanks!
>
> **(b) Hybrid evaluation protocol:** Unlike existing full overlap (e.g., LaSOT) or one-shot (e.g., Got-10k) evaluation protocol, we use a hybrid protocol wherein part of object classes (not videos) in test set have overlap with training set, while the rest classes remains unseen. The reason is that, in real world, humans often track objects from both frequently seen and unseen categories. To develop human-like trackers, we adopt such a hybrid protocol for VastTrack with 561 overlap classes and 141 completely unseen classes in test set.
>
> **(c) Computational resources:** We conduct evaluation experiments on a workstation with an Intel Xeon w9 CPU and 4 Nvidia A6000 GPUs.
>
> Thanks again for these helpful comments and we’ll add the above clarifications in revision.
>
> >**Q4::** The specifics of the data collection process, such as the tools and methods used for video acquisition, could be more detailed.
>
> **A4:** After determining the classes for VastTrack, we start searching for videos from YouTube. For videos that may qualify our task, we record their URLs and starting and end time and then download them in batches using an open-source software YT-DLP (https://github.com/yt-dlp/yt-dlp). We’ll clarify this in revision. Thanks!
>
> >**Q5:** The details of annotation process, the guidelines followed by annotators, and the quality control measures.
>
> **A5:** Thanks for this comment. We compile an annotation team with a few experts working on tracking and a labeling group. We adopt a multi-round strategy for annotation and quality control. Specifically, the initial frame of each video is first labeled by an expert. Afterwards, the labeling group will manually label all other frames. For consistency, a video is processed by the same labeler. After this, to ensure high quality, the annotation will be verified by at least two experts. If the annotation is not unanimously agreed, it will be returned to the same labeler for refinement guided by the feedback from experts until passing verification. We’ll further clarify this in revision. Thanks!
>
> >**Q6:** Include a URL or other access to the dataset.
>
> **A6:** The project webpage (https://github.com/HengLan/VastTrack) in the abstract (Line 21 in paper) contains access to our toolkit, results, and dataset. Please kindly check it out. Thanks.
>
> >**Q7:** About maintaining and updating VastTrack in the future.
>
> **A7:** Thanks for this comment. Our VastTrack is hosted on Github. This allows us to conveniently check feedback from the community and to improve VastTrack via necessary maintenance and updates by research groups of senior authors of VastTrack. Besides, we’ll try our best to continue assessing future trackers to offer up-to-date evaluation and comparison on VastTrack. The ultimate goal is to offer a long-term and stable platform for the tracking community. We’ll clarify this in revision. Thanks!
>
> >**Q8:** Ethical considerations related to data collection.
>
> **A8:** Thanks for this comment. We avoid using private videos in VastTrack and all videos are collected under the Creative Commons license for research only. However, we understand the license might change in future. Once any notification regarding this is received, we’ll take action to handle it. We’ll clarify this in revision. Thanks!
>
> >**Q9:** Guidelines for the responsible use of dataset.
>
> **A9:** Thanks. As suggested, we’ll further clarify the purpose and scope of VastTrack, and its potential bias such as demographic imbalances and limitations and possible solutions to alleviate them in revision.

---

> > ### Comment · Reviewer_5VYv · 2024-08-26
> >
> > I have no more concerns and think the current paper can be accepted.

---

> > > ### Author Response · Authors · 2024-08-26
> > > **Response to reviewer's comment**
> > >
> > > Dear Reviewer,
> > >
> > > Thank you very much for your encouraging comment and feedback on our work. We will improve our work in the final revision as claimed in the rebuttal.
> > >
> > > Again, thanks!

---

### Decision · Program_Chairs · 2024-09-26

**Decision:**

Accept (Poster)

**Comment:**

This paper received 777. The reviewers highlighted the number of categories, scale, quality of annotations, clarity of writing, inclusion of language, and extensive evaluations.

As weaknesses, the short video length and low fps were raised. The paper still presents an advance, and the authors point out they are focussed on short term tracking.

The authors provided a strong rebuttal, including additional results on the requested TNL2K, AQATrack and HIPTrack.